# Distillation Policy Optimization

## Abstract

While on-policy algorithms are known for their stability, they often demand a substantial number of samples. In contrast, off-policy algorithms, which leverage past experiences, are considered sample-efficient but tend to exhibit instability. Can we develop an algorithm that harnesses the benefits of off-policy data while maintaining stable learning? In this paper, we introduce an actor-critic learning framework that harmonizes two data sources for both evaluation and control, facilitating rapid learning and adaptable integration with on-policy algorithms. This framework incorporates variance reduction mechanisms, including a unified advantage estimator (UAE) and a residual baseline, improving the efficacy of both on- and off-policy learning. Our empirical results showcase substantial enhancements in sample efficiency for on-policy algorithms, effectively bridging the gap to the off-policy approaches. It demonstrates the promise of our approach as a novel learning paradigm.

## 1 Introduction

Deep model-free reinforcement learning (RL) has emerged as a promising solution for tackling a wide range of tasks autonomously. Its effectiveness relies on innovations in neural network adaptation, notably the use of replay buffers Lin (1992) Mnih et al. (2013), which helps decorrelate experiences and facilitate more effective weight updates. Off-policy algorithms like DDPG Lillicrap et al. (2016), TD3 Fujimoto et al. (2018), and SAC Haarnoja et al. (2018a) harness these techniques to achieve scalable performance in continuous control tasks. However, off-policy learning encounters the challenge known as the "deadly triad" Sutton & Barto (2018) van Hasselt et al. (2018), which can lead to instability when combining bootstrapping and function approximation. Conversely, on-policy algorithms collect extensive data under the same policy, providing more reliable statistics and greater stability. However, they remain sample-intensive. Recognizing this complementary nature of on-policy and off-policy methods, our objective is to design an algorithm that marries stability with sample efficiency by leveraging the strengths of both approaches.

However, directly applying off-policy techniques to on-policy algorithms can be challenging. In on-policy algorithms, the traditional use of the state value function limits the scope of policy gradients (PG). This contrasts with off-policy algorithms that often employ a broader class of policy gradients, such as the deterministic PG Silver et al. (2014) and the reparameterized PG Haarnoja et al. (2018a). These more versatile PG methods are common in off-policy settings but pose a challenge when adapting to on-policy algorithms. This incompatibility further renders the on-policy techniques such as GAE Schulman et al. (2016) ineffective, which along with the value function as a baseline for variance reduction is crucial for performance. To leverage the advantages of off-policy gradients while retaining the benefits of on-policy methods, we introduce UAE, a unified technique that can accommodate any state-dependent baseline and free the choice of the bootstrapped value. Notably, UAE encompasses GAE as a strict special case. To fully unleash the potential of UAE, we also propose a residual baseline related to the optimal baseline Greensmith et al. (2001) that enhances on-policy gradient estimate. Furthermore, this baseline can be seamlessly integrated into the off-policy gradient, mitigating instability and expediting the learning process. Unlike the traditional value baseline, it exhibits higher sample efficiency, as it is trained entirely off-policy.

With the necessary prerequisites and variance reduction mechanisms in place, our high-level algorithmic design is geared towards fully utilizing off-policy data for both policy evaluation and improvement. In the context of policy evaluation, algorithms often rely on the fitted Q-iteration (FQI) Ernst et al. (2005) Fan

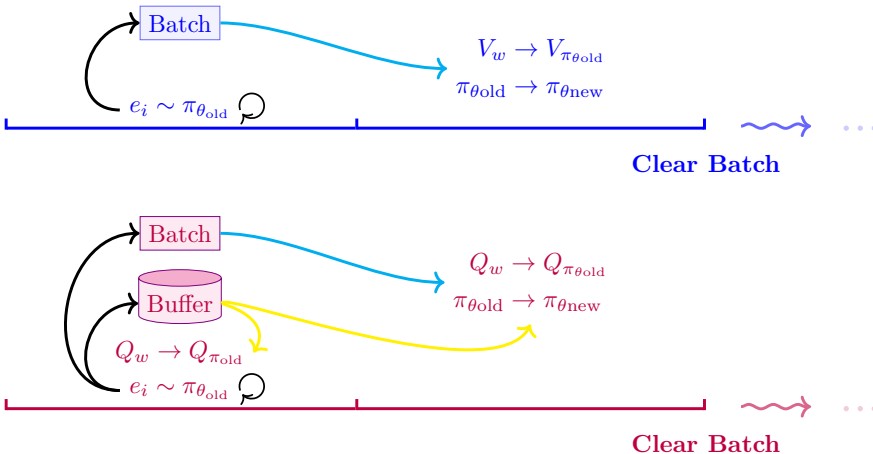

Figure 1: Training procedure comparison between PPO and DPO (Top: PPO; Bottom: DPO), in which $e_i = (s_i, a_i, r_i, s_i')$, generated by interacting with environment, repeating until it reaches a maximum horizon $T$. **Batch** only stores data sampled from the current policy, and will be emptied after a training cycle, while **Buffer** refers to the relay buffer. **Cyan line** indicates data from the source for multiple uses, and **Yellow line** indicates data from the source for individual use. DPO leverages two data sources for both evaluation and control, whereas PPO exclusively relies on the on-policy data.

et al. (2020). This method employs mean squared error loss with stochastic targets. While the Monte Carlo estimate provides unbiased estimates, it can be susceptible to increased variance due to trajectory noise. On the other hand, the use of temporal difference targets, whether deterministic Mnih et al. (2013) or stochastic Haarnoja et al. (2018a), leverages replayed experiences to make predictions, which is amenable to the online learning with high sample efficiency but may introduce bias. We aim to harness the strengths of both approaches to enhance accuracy and generalization. In pursuit of an efficient solution, as illustrated in Figure 1, we adopt a bi-level approach. During the environment interaction, it iteratively updates the critic using replayed experiences. Subsequently, it performs batch updates employing the on-policy data. However, in practice, we have observed that the standard approach, such as FQI, does not perform well at the first level, primarily due to the challenge of long-horizon prediction. To address this issue, we have incorporated distributional learning, which offers a more favorable optimization landscape. To improve the policy, we leverage UAE to estimate the on-policy gradient, which is then interpolated with an optimistic off-policy objective, promoting sample efficiency, stable learning, and inherent exploration.

In this paper, we introduce a general learning framework called Distillation Policy Optimization (DPO), which can be readily applied to various on-policy learners, consistently outperforming its on-policy counterpart and even the state-of-the-art off-policy algorithms on continuous benchmark tasks. Our contributions can be summarized in three main aspects:

- We extend GAE to UAE, offering greater flexibility with different choices of baseline and critic functions.

- We propose a sample-efficient baseline, not only yielding a superior on-policy gradient estimator accompanied with UAE but also effectively facilitating off-policy learning when incorporated into the off-policy gradient.

- We enhance sample efficiency with full engagement of off-policy data through 1) merged policy evaluation with a distributional modeling; 2) interpolated policy gradient with an optimistic objective.

Throughout the paper, we provide comprehensive theoretical insights and empirical results that confirm the effectiveness of DPO, establishing it as a strong competitor in the field.

## 2 Preliminaries

### 2.1 Notation

We consider an infinite-horizon discounted MDP, which formulates how the agent interacts with the environment dynamics. Reinforcement learning aims to solve a sequential problem. Being at the state $s_t \in \mathcal{S}$, the agent takes an action $a_t \in \mathcal{A}$ according to some policy $\pi$, which assigns a probability $\pi(a_t|s_t)$ to the choice. After the environment receives $a_t$, it emits a reward $r_t$, and sends the agent to a new state $s_{t+1} \sim P(s_{t+1}|s_t, a_t)$. Following this procedure, we can collect a trajectory $\tau = (s_0, a_0, s_1, a_1, \dots)$, where $s_0$ is sampled from the distribution of the initial state $\rho_0$. The ultimate goal of the agent is to maximize the expected discounted reward $\eta(\pi) = \mathbb{E}_\tau \left[ \sum_{t=0}^\infty \gamma^t r_t \right]$, with a discount factor $\gamma \in [0, 1)$. We also define the unnormalized discounted state visitation distribution (improper) as $\rho_\pi = \sum_{t=0}^\infty \gamma^t P(s_t = s|\rho_0, \pi)$, and $d_\pi(s, a) = \rho_\pi(s)\pi(a|s)$, corresponding to the state-action one. Whenever noticed, the policy will be parameterized as $\pi_\theta$, sometimes abbreviated as $\pi$ for simplicity's sake. Thus the objective turns out to be $\eta(\theta)$. We declare that we are using the $l^2$-norm variance of a random vector $X$, that is, $\mathbb{V}[X] = \mathbb{E}[\|X - \mathbb{E}[X]\|_2^2]$.

### 2.2 Policy Gradient

The policy gradient Sutton et al. (1999) can be expressed as:

$$\nabla_\theta \eta(\theta) = \mathbb{E}_{s \sim \rho_{\pi_\theta}, a \sim \pi_\theta} \left[ \nabla_\theta \log \pi_\theta(a|s) Q^{\pi_\theta}(s, a) \right]. \tag{1}$$

In practice, incorporating a state-dependent baseline function $b(s)$ can not only reduce the variance drastically but also not intervene with the expectation Greensmith et al. (2001). Combine it, we have:

$$\begin{aligned} \nabla_\theta \eta(\theta) &= \mathbb{E}_{s \sim \rho_{\pi_\theta}, a \sim \pi_\theta} \left[ \nabla_\theta \log \pi_\theta(a|s)(Q^{\pi_\theta}(s, a) - b(s)) \right] \\ &= \mathbb{E}_{s \sim \rho_{\pi_\theta}, a \sim \pi_\theta} \left[ \nabla_\theta \log \pi_\theta(a|s) A^{\pi_\theta, b}(s, a) \right]. \end{aligned} \tag{2}$$

### 2.3 Optimal Baseline

The optimal baseline can be derived by obtaining the fixed point of the variance of Equation 2:

$$b^\star(s) = \frac{\mathbb{E}_{\pi_\theta}[u_\theta(s, a)^\top u_\theta(s, a) Q^{\pi_\theta}(s, a)|s]}{\mathbb{E}_{\pi_\theta}[u_\theta(s, a)^\top u_\theta(s, a)|s]}, \tag{3}$$

where $u_\theta(s, a) = \nabla_\theta \log \pi_\theta(a|s)$. Its derivation can be found in Appendix A.

However, this baseline is rarely used in practice, because it is extremely demanding for computing the $u_\theta(s_t, a_t)$ for each time step of the available data.

### 2.4 Distributional Reinforcement Learning

Distributional reinforcement learning Bellemare et al. (2017) abstracts the appraisal $Q^\pi(s, a)$ as a distribution $Z^\pi(s, a)$, whose expectation corresponds to the actual value of $Q$. In this perspective, the Bellman expectation operator is reloaded as:

$$\begin{aligned} \mathcal{T}^\pi Z(s, a) &\overset{\mathrm{D}}{:=} r(s, a) + \gamma Z(s', a') \\ s' &\sim P(\cdot|s, a), a' \sim \pi(\cdot|s'), \end{aligned} \tag{4}$$

where equality is held under probability laws.

## 3 Unified Advantage Estimator

To combat the noise arising from the long-delayed signals, GAE reduces the temporal spread by letting $b(s) = V(s)$ and then shrinks the long-term effect with a steeper parameter $\lambda$. Let $\delta_t^V = r_t + \gamma V(s_{t+1}) - V(s_t)$,

we have:

$$\hat{A}_t^{\mathrm{GAE}(\gamma,\lambda)} = \sum_{l=0}^{\infty} (\gamma\lambda)^l \delta_{t+l}^V. \tag{5}$$

One major limitation is that we don't have more flexible choices except we can stick both of the bootstrapped value function and the baseline function to the value function $V$. This restricted applicability makes it difficult to improve the baseline function for further variance reduction and miss out on a potentially broader class of policy gradients that rely on the state-action value function. We would therefore aim to relax it for both parts. Analogously, we can define a new TD residual $\delta_t = r_t + \gamma\Psi_{t+1} - b(s_t)$, where the bootstrapped value function $\Psi_{t+1}$ can either be $Q(s_{t+1}, a_{t+1})$ or $V(s_{t+1})$, and $b(s)$ is an arbitrary state-dependent function. The core idea is that we can introduce a correction term $z_t = \Psi_t - b(s_t)$ to make the $n$-step estimator unbiased when the true value of $\Psi_t$ is attained

**Proposition 3.1.** *For any $n \in \mathbb{N}^+$, $A_t^{(n)}$ is an unbiased estimator of $A_t^{\pi,b}$, where*

$$\begin{aligned}
A_t^{(n)} &= \delta_t + \sum_{l=1}^{n-1} \gamma^l (\delta_{t+l} - z_{t+l}) \\
&= r_t + \gamma r_{t+1} + \cdots + \gamma^{n-1} r_{t+n-1} + \gamma^n \Psi_{t+n} - b_t.
\end{aligned} \tag{6}$$

Be that as it may, in practice, $\Psi_t$ is the approximate value and thus the $A_t^{(n)}$ is referenced as $\hat{A}_t^{(n)}$. Similarly, we can introduce a steeper parameter $\lambda$ to shrink the long-term effect, by telescoping on which, we would arrive at the unified advantage estimator (UAE):

$$\hat{A}_t^{\mathrm{UAE}(\gamma,\lambda)} = \delta_t + \sum_{l=1}^{\infty} (\gamma\lambda)^l (\delta_{t+l} - z_{t+l}). \tag{7}$$

The intuition behind this estimator is that we correct any TD residual term one step beyond $t$ to the TD error $\delta_{t+l} - z_{t+l} = r_{t+l} + \gamma\Psi_{t+l+1} - \Psi_{t+l}, l \geq 1$, and leave the first term $\delta_t = r_t + \gamma\Psi_{t+1} - b_t$ to have a potential lower variance dependent on $b_t$. It is worth noting that if we set $\Psi_{t+1} = V(s_{t+1})$ and $b(s_t) = V(s_t)$, then we have GAE exactly. However, its usefulness extends beyond that, as we have the flexibility to choose any state-dependent baseline and extend it to the state-action value function. When combined with a high-quality baseline, it can effectively reduce both the instantaneous variance resulting from sampling from the state and policy distribution, as well as the variance arising from sampling a trajectory $\tau$. A truncated version of such an estimator is summarized in Algorithm 1.

---

**Algorithm 1** Unified Advantage Estimator

**Input:** $\gamma$, $\lambda$, Batch size $T$, rewards $r$, Q values $Q$, baselines $b$, dones $d$
**Initialize** uae $= 0$
**for** $t = T - 1, T - 2, \ldots, 0$ **do**
    $\delta = r_t + \gamma Q_{t+1}(1 - d_{t+1}) - b_t$
    $z = Q_t - b_t$
    discounted uae $= \gamma\lambda(1 - d_{t+1})$uae
    $A_t = \delta +$ discounted uae
    uae $= (\delta - z) +$ discounted uae
**end for**
**return** advantages $A$

---

**Connection to SARSA($\lambda$)** TD($\lambda$) updates the value function towards the $\lambda$-return $G_t^{\lambda,V}$, for which a useful identity is often used to establish the connection between the forward- and backward-view Sutton (1988) Sutton & Barto (2018):

$$\begin{aligned}
G_t^{\lambda,V} &= (1 - \lambda) \sum_{n=1}^{\infty} \lambda^{n-1} G_t^{(n),V} \\
&= V(s_t) + \sum_{n=0}^{\infty} (\gamma\lambda)^n \delta_{t+n}^V,
\end{aligned} \tag{8}$$

where $G_t^{(n),V} = \sum_{k=0}^{n-1} \gamma^k r_{t+k} + \gamma^n V(s_{t+n})$ is the $n$-step return.

It is evident that GAE is a variance-reduced $\lambda$-return with a baseline $V$, represented as $\hat{A}_t^{\mathrm{GAE}(\gamma,\lambda)} = G_t^{\lambda,V} - V_t$. Similarly, we can interpret the UAE in the same way – UAE is a variance-reduced $\lambda$-return with an arbitrary baseline $b$, denoted as $\hat{A}_t^{\mathrm{UAE}(\gamma,\lambda)} = G_t^{\lambda,Q} - b_t$, since this identity also holds for SARSA($\lambda$) (see Appendix B.3 for derivation).

To fully exploit the potential of UAE, in the next section, our goal is to learn a more adaptable baseline that can be seamlessly integrated into the off-policy gradient at a later stage.

## 4  Residual Baseline

The optimal baseline (Equation 10) is essential as it offers the most significant reduction in policy gradient variance, thereby maximizing performance gains.. However, in practice, obtaining the $u_\theta(s,a)$ can be computationally demanding, especially when dealing with a large batch of data. Consequently, achieving the optimal baseline becomes challenging. Moreover, in practical scenarios, the value function used as a baseline does not accurately capture its true value due to estimation errors Ilyas et al. (2020). We aim to alleviate the computational overhead and enhance the quality of the baseline. By observing the structure of the optimal baseline, we define:

$$\tilde{\pi}(a|s) = \frac{\pi(a|s)u_\theta^\top u_\theta}{\mathbb{E}_\pi[u_\theta^\top u_\theta|s]}, \qquad l(s,a) = \frac{u_\theta^\top u_\theta}{\mathbb{E}_\pi[u_\theta^\top u_\theta|s]}. \tag{9}$$

Then we can rewrite the optimal baseline as:

$$b^\star(s) = \mathbb{E}_{\tilde{\pi}}[Q^\pi(s,a)|s]. \tag{10}$$

Using the importance sampling, we have:

$$\begin{aligned} b^\star(s) &= \mathbb{E}_\pi\big[\frac{\tilde{\pi}(a|s)}{\pi(a|s)}Q^\pi(s,a)|s\big] \\ &= \mathbb{E}_\pi[l(s,a)Q^\pi(s,a)|s]. \end{aligned} \tag{11}$$

We would directly parameterize the $l(s,a)$, or, alternatively, introduce a "residual" term $\mathrm{r}_\phi(s,a)$ to reformulate the $l(s,a)$ as $1 + \mathrm{r}_\phi(s,a)$, since $\mathbb{E}_\pi[l(s,a)] = 1$. This transformation would induce a symmetric behavior for $\mathrm{r}_\phi(s,a)$, which is favored by the neural network. In the hope that $Q_w$ is a good approximation to $Q^\pi$, it translates to approximate the Equation 11 as:

$$b_\phi^\pi(s) = \mathbb{E}_\pi[(1 + \mathrm{r}_\phi(s,a))Q_w(s,a)|s]. \tag{12}$$

In practice, we can sample $m$ actions $a_1, a_2, \ldots, a_m$ from the $\pi(\cdot|s)$ to approximate the outer expectation.

We then construct a magnitude-free objective to represent the amount of the variance associated with the approximate baseline, which is more tractable and easier to optimize Gu et al. (2017b) Mnih & Gregor (2014):

$$\mathcal{J}(\phi) = \mathbb{E}_{d_\beta}[(Q_w(s,a) - b_\phi^\pi(s))^2], \tag{13}$$

where $d_\beta$ is a mixture of the joint distributions of the past policy sequences. This objective has a wider coverage of past experiences than solely relying on the on-policy data, which is critical to reducing the variance of both on- and off-policy gradient. It is this reason that allows us to make the most of the advantages of the two kinds.

## 5  Practical Algorithm

In this section, we propose a sample-efficient algorithm that combines on- and off-policy data, leveraging the variance reduction techniques discussed earlier. Our approach begins with policy evaluation, updating

the critic not only during the environment interaction using replayed experiences but also through batch updates. Then we interpolate the on-policy gradient with an optimistic objective that incorporates the residual baseline for stable and efficient learning.

## 5.1 Policy Evaluation: Distributional Regression

To being with, we delve into the details of policy evaluation, which employs a bi-level optimization, merging both on- and off-policy updates. It integrates with distributional learning[1]—a strategic response to challenges observed, notably the long-horizon prediction issue encountered with standard approaches like FQI.

The distributional perspective enriches predictions by incorporating input-dependent variance to accommodate uncertainty. It can penalize high-noise input regions and enhance representation learning Shahriari et al. (2022). Leveraging this representation, we develop an efficient update scheme based on the analytical Gaussian modeling, conducted during the environment interaction using the off-policy data. This scheme, akin to the FQI used in TD3 or SAC, shares the same time complexity but does not require excessive samples from the distributional critic (see Appendix D.3). The general update rule can be expressed as:

$$\mathcal{J}(w) = \mathbb{E}_{(s,a,r,s')\sim\mathcal{D}}[D_{\mathrm{KL}}(r + \gamma Z_{\bar{w}}(s', a')||Z_w(s, a))], \tag{14}$$

where $a' \sim \pi(\cdot|s')$, and $Z_{\bar{w}}$ is the target network, commonly used in the off-policy learning to stabilize the neural network.

Once the rollouts are collected, we sample multiple instances from the distributional critic to create a vector $\vec{Z}$ of length $l$ for each $(s, a)$ in the batch $\mathcal{B}$. We then compute a collection of advantages by UAE as $\vec{A}$ and replenish the baseline to construct a target vector $\vec{U} = \vec{A} + \mathbf{1} \cdot b$, where $\mathbf{1}$ is an all-one vector of length $l$. As the entropy term in the KL divergence does not provide gradient information, we minimize the empirical cross-entropy as follows:

$$\hat{\mathcal{J}}(w) = \mathbb{E}_{(s,a)\sim\mathcal{B}}[-\frac{1}{l}\mathbf{1}^\top \log P_w(\vec{U}|s, a)]. \tag{15}$$

This approach effectively harnesses the benefits of both off-policy learning's high sample efficiency and on-policy learning's informative target estimates. Additionally, the estimated advantage is a key component of the on-policy gradient, as discussed later.

## 5.2 Policy Improvement: Advantageous Interpolation

We aim to integrate the off-policy policy gradient with the on-policy policy gradient to enable faster learning, boost sample efficiency, and encourage exploration. One solution for this integration involves introducing an interpolating parameter $w \in [0, 1]$ to directly adjust both gradients, as suggested by IPG Gu et al. (2017a):

$$\omega\mathbb{E}_{\rho^{\pi_\theta}, \pi_\theta}[A^{\pi_\theta, V}] + (1 - \omega)\mathbb{E}_{\rho^\beta, \pi_\theta}[Q_w]. \tag{16}$$

While approximating a value function $V$ for advantage calculation, it also maintains an off-policy critic $Q_w$. This separate estimation can pose a problem as it leads to varying magnitudes between the two types of policy gradients. Even when working with off-policy data, the estimation of $Q_w$ can still be prone to errors and noise. Therefore, we incorporate the residual baseline to mitigate the noise and only update actions that provide an advantage. This approach can be efficiently optimized with trust region methods:

$$\mathcal{J}(\theta) = \omega\mathbb{E}_{\rho^{\pi_{\theta_{\mathrm{old}}}}, \pi_\theta}[A^{\pi_{\theta_{\mathrm{old}}}, b_\phi^{\pi_{\theta_{\mathrm{old}}}}}] + (1 - \omega)\mathbb{E}_{\rho^\beta, \pi_\theta}[A^+ - \alpha \log \pi_\theta(a|s)], \tag{17}$$

where $A^+ = (Q_w - b_\phi^{\pi_\theta})^+$ is the positive advantage, of which $(x)^+$ stands for $\max(x, 0)$. To enhance exploration and prevent premature convergence, we include an entropy bonus controlled by parameter $\alpha$ Mnih et al. (2016). This bonus improves the exploration of the off-policy data. While our off-policy gradient resembles the likelihood ratio gradient estimator of SAC, the purpose of the entropy term in our approach differs. It is not considered as part of the task-specific reward for the agent.

---

[1]The procedure is generally applicable to other alternatives in place of the distributional loss.

---

**Algorithm 2** Distillation Policy Optimization

---

**Initialize parameters** $w, \bar{w}, \theta, \phi$
**for** each iteration **do**
    **for** each environment step **do**
        Execute $a_t \sim \pi(\cdot|s_t)$, observe reward $r_t$ and next state $s_{t+1}$
        Store transition $(s_t, a_t, r_t, s_{t+1})$ to the replay buffer $\mathcal{D}$ and the batch $\mathcal{B}$
        Sample mini-batch of $n$ transitions $(s, a, r, s')$ from $\mathcal{D}$
        Update critic with $\nabla_w \mathcal{J}(w)$ (Equation 14)
        Update target network $\bar{w} \leftarrow \tau w + (1 - \tau)\bar{w}$
    **end for**
    **for** each baseline update **do**
        Sample mini-batch of $n$ transitions $(s, a, r, s')$ from $\mathcal{D}$
        Update baseline with $\nabla_\phi \mathcal{J}(\phi)$ (Equation 13)
    **end for**
    Calculate $\hat{A}$ for $\mathcal{B}$ by Algorithm 1
    **for** each epoch **do**
        **for** each policy update **do**
            Sample mini-batch from $\mathcal{B}$, and compute $\mathcal{J}_{\text{on-policy}}(\theta)$
            Sample mini-batch of $n$ transitions $(s, a, r, s')$ from $\mathcal{D}$, and compute $\mathcal{J}_{\text{off-policy}}(\theta)$
            Update policy with $\nabla_\theta \mathcal{J}(\theta)$ (Equation 17)
            Update critic with $\nabla_w \hat{\mathcal{J}}(w)$ (Equation 15)
            Update target network $\bar{w} \leftarrow \tau w + (1 - \tau)\bar{w}$
        **end for**
    **end for**
    Reset the batch $\mathcal{B}$
**end for**

---

Without the cancellation of the negative part, when an action is perceived as unfavorable, it steers away from that choice and increases the likelihood of exploring unknown actions. This introduces a risk of making completely wrong decisions. However, the process we employ, which eliminates negative aspects and emphasizes positive signals through the residual baseline, prevents detrimental updates and ensures that the movement direction is always advantageous.

# 6 Theoretical Analysis

In this section, we present a theoretical analysis of the proposed methods. We explore three key questions: **(1)** How does UAE outperform GAE, and what's their relationship? **(2)** Can the residual baseline effectively minimize variance in the off-policy gradient? **(3)** What advantages does our interpolated policy gradient offer? These questions form the foundation of our theoretical investigation.

**Assumption 6.1.** $\sup_{s,a} |Q_w|$ is bounded by some constant $M$.

**Assumption 6.2.** $\sup_{s,a} \|\nabla_\phi r_\phi\|$ is bounded by some constant $G$.

**Assumption 6.3.** $\sup_{s,a} |r_\phi|$ is bounded by some constant $K$.

We begin by examining the relationship between UAE and GAE in terms of the variance of the on-policy gradient.

**Theorem 6.4.** *For any choice of $\Psi$ being either $Q^\pi$ or $V^\pi$, and $b$ state-dependent, then*

$$\mathbb{V}_{s_t,a_t}[u_\theta A_t^{UAE(\gamma,\lambda)}] - \mathbb{V}_{s_t,a_t}[u_\theta A_t^{GAE(\gamma,\lambda)}] = \overbrace{\mathbb{E}_{s_t,a_t}\left[u_\theta^\top u_\theta\left(\sum_{l=0}^\infty (\gamma\lambda)^{2l} \cdot \gamma^2(1-\lambda^2)\mathbb{E}_{s_{t+l+1},a_{t+l+1}}\left[(\Psi - V^\pi)^2\right]\right)\right]}^{irreducible} +$$

$$\underbrace{\mathbb{E}_{s_t,a_t}[u_\theta^\top u_\theta(b^2 - V^{\pi 2} - 2Q^\pi(b - V^\pi))]}_{reducible}.$$

$$(18)$$

Notably, if $\Psi = Q^\pi$, an irreducible error arises from the additional action in $Q^\pi(s, a)$. In the case of episodic tasks, it provides insights that bootstrapping from $Q^\pi$ introduces additional complexity when $\lambda$ is low, but it becomes eliminable when full return is achieved. With $\Psi = V^\pi$, the problem simplifies, allowing for a direct comparison.

**Corollary 6.5.** *If $\Psi = V^\pi$, for any baseline $b(s)$ that reduces variance no less than $V^\pi(s)$, then*

$$\mathbb{V}_{s_t, a_t}[u_\theta A_t^{UAE(\gamma,\lambda)}] \leq \mathbb{V}_{s_t, a_t}[u_\theta A_t^{GAE(\gamma,\lambda)}]. \tag{19}$$

UAE's advantage over GAE is its capacity to broaden baseline options, notably enhancing variance reduction with improved baselines beyond $V^\pi$. This underscores the crucial role of baseline selection in unlocking UAE's full potential.

We will now delve into the understanding of the residual baseline concerning data exposure and its application for the off-policy gradient.

**Theorem 6.6.** *Under Assumption 6.2, at nth iteration, let $k = \min\{n, \lfloor \frac{|\mathcal{D}|}{T} \rfloor\}$, for the current policy $\pi_n$, along with its predecessors $\{\pi_{n-i}\}_{i=1}^{k-1}$, if for any $s \in \mathcal{S}$, $\sup_i D_{TV}(\frac{d_{n-i}(s,a)}{d_\beta(s)} || \pi) < \frac{\epsilon}{4}$ and $\mathbb{E}_\pi[|r_{\phi^\star}(s,a)|] < \frac{\epsilon}{2}$, then $\|\nabla_\phi \mathcal{J}(\phi^\star)\| < 2GM^2 \epsilon$.*

This suggests that optimizing the residual baseline may become increasingly challenging as the volume of its training data grows. Therefore, in practice, selecting an appropriate size for training baseline is of vital importance.

**Corollary 6.7.** *Under Assumption 6.3, for any successor policy $\tilde{\pi}$ of $\pi$, if $\sup_s D_{KL}(\tilde{\pi} || \pi) < \frac{\epsilon}{2}$, then*

$$\mathbb{E}_{d_\beta}[(Q_w(s,a) - b_{\phi^\star}^{\tilde{\pi}}(s))^2] \leq 2\mathcal{J}(\phi^\star) + 2((K+1)M)^2 \epsilon. \tag{20}$$

Following the optimization of the residual baseline, its integration with trust region methods ensures effective maintenance of low magnitude-free variance in evolving policies. This, in turn, promotes stable parameter adjustments, ultimately leading to a smoother learning process.

Regarding our interpolated policy gradient, its advantage lies in two folds: self-annealing effect and potential for a tighter lower bound.

**Theorem 6.8.** *(Self-annealing effect) Under Assumption 6.3, for any policy sequence $\{\pi_k\}$ such that its limiting point $\pi^\star$ lies in the deterministic optimal policy set, if for any $s \in \mathcal{S}$, $\lim_{k\to\infty} \mathbb{E}_{\pi_k}[r_{\phi_k}] = 0$, then*

$$\limsup_{k\to\infty} A_k^+ = 0. \tag{21}$$

It indicates that as the positive advantage diminishes, the surrogate reduces to encourage exploration only. This self-annealing effect is helpful since as the learning evolves, the direction of the policy update will close to the on-policy gradient, which is generally stabler.

**Theorem 6.9.** *(Bounded bias) Let $\Delta = \max_{s,a} |Q^\pi - Q_w|$, $\Omega = \max_s |\mathbb{E}_{\tilde{\pi}}[Q_w - b_\phi^{\tilde{\pi}}]|$, $\Upsilon = \max_s |\mathbb{E}_{\tilde{\pi}}[Q^\pi - b_\phi^\pi]|$, and define*

$$L_\pi(\tilde{\pi}) = \eta(\pi) + \omega \mathbb{E}_{\rho^\pi, \tilde{\pi}}[Q^\pi - b_\phi^\pi] + (1-\omega)\mathbb{E}_{\rho^\beta, \tilde{\pi}}[(Q_w - b_\phi^{\tilde{\pi}})^+ - \alpha \log \tilde{\pi}], \tag{22}$$

*then*

$$|\eta(\tilde{\pi}) - L_\pi(\tilde{\pi})| \leq \frac{2\gamma\Upsilon}{(1-\gamma)^2}\sqrt{D_{KL}^{max}(\pi || \tilde{\pi})} + (1-\omega)(\Delta + C_1\sqrt{2D_{KL}^{max}(\pi || \tilde{\pi})} + \frac{2\gamma\Omega}{(1-\gamma)^2}\sqrt{D_{KL}^{max}(\pi || \beta)} + C_2). \tag{23}$$

This theorem provides a comprehensive bound on the bias introduced by the on-policy state distribution mismatch and the off-policy learning, from $L_\pi(\tilde{\pi})$ to the true objective $\eta(\tilde{\pi})$. The accuracy of this approximation depends on the deviation from the original policy $\pi$, the approximation quality of $Q_w$ to $Q^\pi$, and the extent of off-policyness. Our combined policy evaluation greatly reduces the $\Delta$ gap. In the case where $r \equiv 0$, it further eliminates bias from off-policy learning as $\Omega = 0$ for any $\tilde{\pi}$. IPG, on the other hand, fits $Q_w$ using only off-policy data and struggles to manage off-policyness as their $\Omega$ is policy-dependent. While the residual term is typically non-zero, it remains relatively small (see Appendix I.4, Figure 10(a)), reducing off-policyness. When combined with any on-policy gradient with an enforced trust region, it can effectively

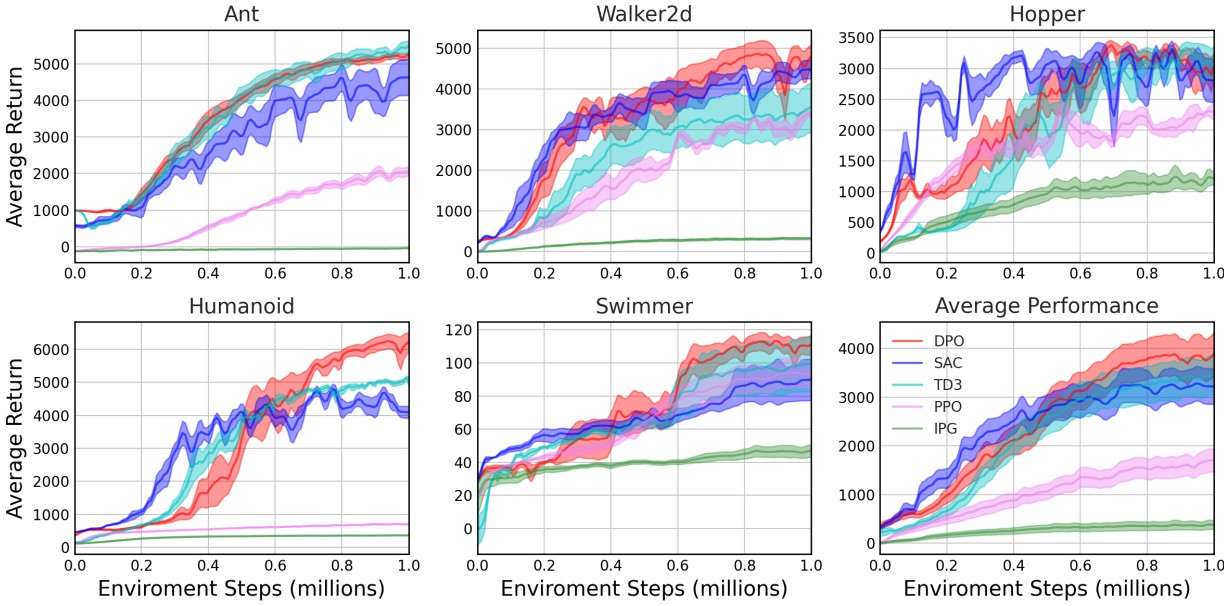

Figure 2: Learning curves on continuous control tasks, averaged over 5 random seeds and shaded with standard error.

utilize recent rollouts while constraining the deviation from the origin. The shifting constant $C_2$ can also be controlled, depending on the desired level of exploration and the portion of the negative off-policy gradient to be canceled[2].

## 7 Experiments

Our goal is to validate sample efficiency and stable learning while understanding the contributions of different algorithmic components. We perform our algorithm on several continuous control tasks from the OpenAI Gym Brockman et al. (2016) with the MuJoCo simulator Todorov et al. (2012).

**Evaluation** Since our algorithm is in a hybrid fashion, the policy that we update would not strictly follow the sampling policy. We thus evaluate our algorithm by executing the mean action with 10 trails, for which we report the averaged episodic reward every 4096 steps. We run each task with 5 random seeds, whose total environment step is 1 million.

As our default on-policy learner is PPO, it is a direct baseline to verify whether our method realizes an improvement. And we test the advantage of the unified learning of the critic and the optimistic policy gradient against IPG. We also made comparisons with the state-of-the-art off-policy algorithms, such as SAC Haarnoja et al. (2018a) and TD3 Fujimoto et al. (2018). We defer additional comparisons to related baselines that combine on-policy methods with off-policy data to Appendix I.2.

The learning curves are presented in the Figure 2. DPO demonstrates superior or comparable performance across all tasks, notably excelling in the high-dimensional Humanoid task. Other DPO variants, such as DPO(A2C) and DPO(TRPO), exhibit significant improvements over their on-policy counterparts (Table 1). This highlights the potential of our method as a promising learning paradigm for a range of on-policy algorithms.

---

[2]Although we mainly focus on the negative portion, the bound is generally applicable to any removed portion.

**Sample Efficiency** DPO combines both on- and off-policy evaluation and employs off-policy gradient interpolation to enhance sample efficiency. As depicted in Figure 3(a), DPO achieves exceptional performance more rapidly and with significantly less time compared to off-policy algorithms. To achieve comparable performance, on-policy algorithms like PPO necessitate 10 times more samples than DPO. This underscores DPO's enhanced sample efficiency over on-policy algorithms and improved time efficiency compared to off-policy algorithms. Additionally, DPO only performs 4% of the total number of policy gradients, compared to off-policy algorithms like TD3 and SAC, highlighting DPO's superior data utilization per update.

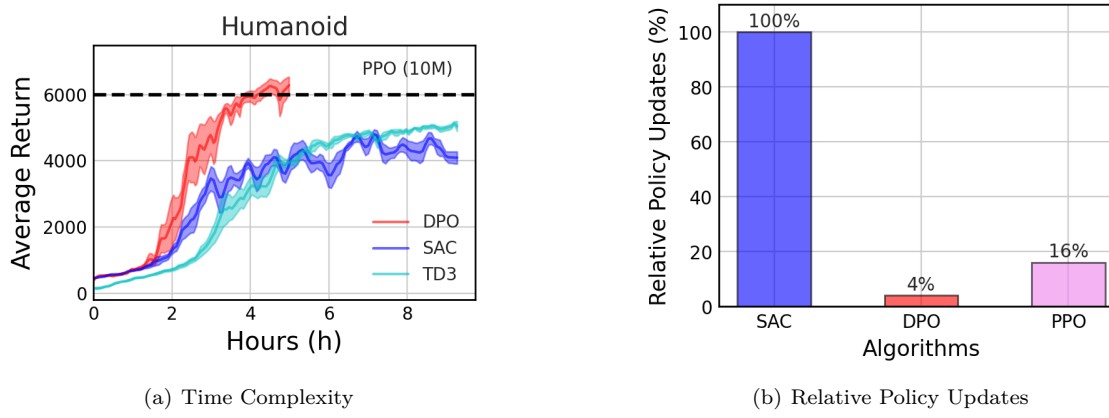

(a) Time Complexity                    (b) Relative Policy Updates

Figure 3: Comparison of time complexity and relative parameter updates.

**Stable Learning** DPO preserves the stable learning characteristics typically associated with on-policy algorithms. We assess stability by measuring the variability in policy changes and critic mean squared error (MSE) loss. These metrics are computed using the variance of parameter updates and the average total variation respectively (see definitions in the Appendix F.3). Both metrics are calculated based on 25% of the data within fixed intervals. To ensure a fair comparison, we normalize the losses before calculating the latter metric, as loss functions can vary in magnitude for different algorithms. Figure 4(a) shows that DPO maintains smoother policy changes compared to SAC, even without the use of learning rate scheduling as seen in PPO. Furthermore, Figure 4(b) highlights that DPO exhibits reduced variability in critic MSE loss, suggesting the effectiveness of the combined policy evaluation within a more stable optimization landscape.

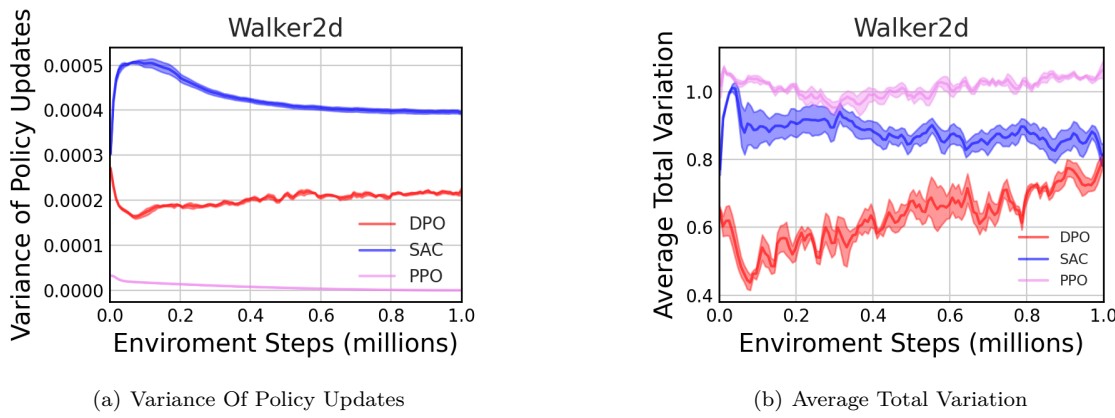

(a) Variance Of Policy Updates              (b) Average Total Variation

Figure 4: Comparison of variance of policy updates and average total variation of critic MSE loss.

**General Framework** One of our key innovations is the provision of the implementation for a versatile learning framework that can adapt to various on-policy gradient estimators, including A2C, TRPO, and

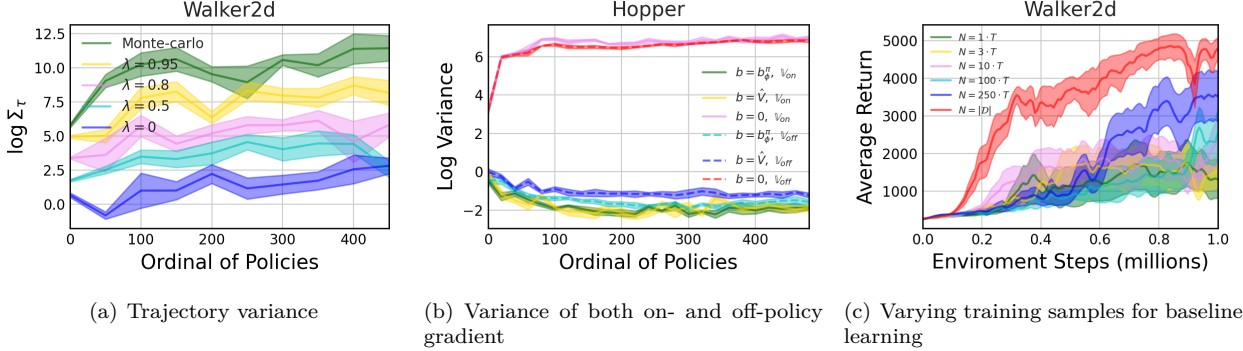

Figure 5: Variance reduction of UAE and residual baseline, and data exposure of baseline learning.

PPO (details can be found in Appendix F.8). Within this framework, these estimators share the same set of hyperparameters across different tasks, except for any algorithmic specifications. The remarkable improvement in sample efficiency is evident in Table 1 and Figure 2, enabling on-policy algorithms to successfully tackle previously unsolvable tasks. It also narrows the performance gap with off-policy approaches, while increasing time efficiency.

Table 1: Head-to-head comparison between other variants of DPO to its counterpart.

| METHOD | WALKER2D | HOPPER | SWIMMER | ANT | HUMANOID | AVG. |
|---|---|---|---|---|---|---|
| A2C | $134 \pm 45$ | $156 \pm 30$ | $17 \pm 4$ | $942 \pm 3$ | $163 \pm 50$ | $282 \pm 26$ |
| DPO(A2C) | $\mathbf{2786} \pm 681$ | $\mathbf{2108} \pm 712$ | $\mathbf{40} \pm 5$ | $\mathbf{3581} \pm 905$ | $\mathbf{3980} \pm 2231$ | $\mathbf{2499} \pm 907$ |
| TRPO | $2449 \pm 251$ | $\mathbf{2142} \pm 591$ | $\mathbf{103} \pm 21$ | $68 \pm 50$ | $503 \pm 23$ | $853 \pm 187$ |
| DPO(TRPO) | $\mathbf{3581} \pm 516$ | $2025 \pm 1120$ | $55 \pm 24$ | $\mathbf{4615} \pm 129$ | $\mathbf{5011} \pm 1197$ | $\mathbf{3057} \pm 597$ |

**Variance Reduction** We explore two methods for variance reduction: UAE and a residual baseline. We investigate three key questions: **(1)** Does UAE effectively mitigate long-term noise? **(2)** Does the residual baseline reduce both on-policy and off-policy gradient variance? **(3)** How does the baseline's data exposure impact its performance?

For the first question, we evaluate the variance of the policy gradient equipped with UAE using the law of total variance, as expressed in Equation 24. Our focus is primarily on $\Sigma_\tau$, which arises from trajectory sampling. We investigate how varying the value of $\lambda$ allows us to reduce the temporal spread and mitigate the trajectory noise, as depicted in Figure 5(a).

$$\mathbb{V}_{s,a}[u_\theta A^{\mathrm{UAE}(\gamma,\lambda)}] = \mathbb{E}_{s,a}[\mathbb{V}_{\tau|s,a}[u_\theta A^{\mathrm{UAE}(\gamma,\lambda)}]] + \mathbb{V}_{s,a}[\mathbb{E}_{\tau|s,a}[u_\theta A^{\mathrm{UAE}(\gamma,\lambda)}]]$$
$$= \Sigma_\tau + \Sigma_{s,a}. \tag{24}$$

Regarding the second question, we compare our residual baseline $b_\phi^\pi(s)$ with an approximate value baseline $\hat{V}(s)$, the sample mean of $Q_w(s, a_i)$ (with $a_i \sim \pi$), and a zero baseline. The results demonstrate the significance of incorporating a baseline. The residual baseline reduces off-policy variance more effectively while maintaining a similar reduction in on-policy variance (see Figure 5(b)).

To address the third question, we examine how the baseline's exposure to data impacts its effectiveness. Gradually increasing the number of segments reveals that training the baseline with more data enhances its ability to stabilize the learning process (see Figure 5(c)).

**Distributional Critic** DPO proactively performs off-policy evaluation steps before the policy improvement starts. In such a long-horizon scenario (as seen in Figure 1), it requires preventing overfitting on the finite dataset and generalizing well on the unseen data. We empirically test this ability on our method along

with other predominant evaluation methods, such as MSE with single Q Ernst et al. (2005) Fan et al. (2020), and MSE with double Q Fujimoto et al. (2018). The results in Figure 6 demonstrate that the distributional critic in DPO excels in both online prediction and training generalization.

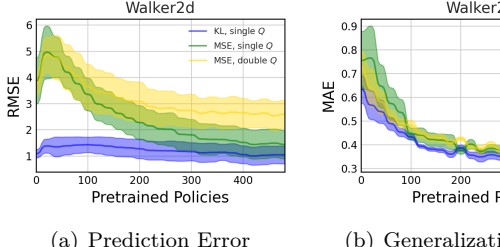

(a) Prediction Error    (b) Generalization Error

Figure 6: **Left**: root mean squared error (RMSE) between $\mathbb{E}[r + \gamma Z_w(s', a')]$ and $\mathbb{E}[Z_w(s, a)]$ on the on-policy rollouts; **Right**: mean absolute error (MAE) between the true value $Q^\pi$ and the fitted value $Q_w$ on the test data.

Table 2: Ablation study on key components, policy gradient, and data usage.

| METHOD | WALKER2D | HOPPER | ANT |
|---|---|---|---|
| NO-UAE | $3710 \pm 562$ | $2436 \pm 1282$ | $1747 \pm 227$ |
| NO-RB | $4128 \pm 807$ | $1090 \pm 213$ | $3866 \pm 1251$ |
| NO-DIST | $4408 \pm 388$ | $2988 \pm 239$ | $3961 \pm 921$ |
| NO-INT | $3341 \pm 1718$ | $2800 \pm 1030$ | $3560 \pm 1775$ |
| NO-ENT | $4194 \pm 790$ | $2830 \pm 910$ | $4095 \pm 1175$ |
| ONLY-ON | $4174 \pm 980$ | $2601 \pm 1107$ | $4053 \pm 814$ |
| ONLY-OFF | $3743 \pm 1186$ | $\mathbf{3223} \pm 329$ | $3857 \pm 1083$ |
| DPO | $\mathbf{4860} \pm 680$ | $3187 \pm 351$ | $\mathbf{5278} \pm 173$ |

**Ablation Study** We analyze the contributions of UAE, the residual baseline, and distributional learning to the performance improvement achieved by DPO. The results are presented in the first group of Table 2. Removing these components leads to varying degrees of performance decrease, underscoring the effectiveness of each component. In the second group, when we eliminate interpolation, both entropy regularization and positive advantage are removed, resulting in inferior performance. Retaining the positive advantage but removing the entropy regularization leads to improved results but still falls short of comparable performance. The results in the third group demonstrate that combining on- and off-policy evaluation is essential for realizing the full benefits of distributional learning. These findings highlight how DPO enhances sample efficiency, promotes exploration, and enables stable learning.

## 8 Related Work

**Variance reduction techniques** State-dependent baseline has been widely studied in Greensmith et al. (2001) Weaver & Tao (2001), whose application can be found in Peters & Schaal (2008). Although the optimal baseline sounds from a theoretical perspective, its practical use is rare. The conventionalized alternative – the value function that can be estimated directly from the interaction, prevails in on-policy algorithms, such as REINFORCE Williams (1992) and A2C Sutton & Barto (2018). However, it is often brittle to the quality of the function approximation Ilyas et al. (2020) and has a gap between the optimal one. There are also fruitful works that focus on the action-dependent baseline, with stein identity Liu et al. (2018), or factorization Wu et al. (2018). But Tucker et al. (2018) points out that the gain of the action-dependent baseline is often insignificant due to the function approximation and overweighed by other variance components such as the trajectory variance. GAE Schulman et al. (2016) accounts for this by exponentially interpolating different advantage estimators to reduce the temporal spread, while introducing some bias. In Monte Carlo theory, the baseline methods are different kinds of control variate. Beyond RL, to reduce variance, gradient-based optimization can combine both control variate and the reparameterization estimator Grathwohl et al. (2018), and inference problems Mnih & Gregor (2014) Paisley et al. (2012) utilize the control variate for score function estimator.

**Distributional learning** The distributional perspective in reinforcement learning (RL) extends scalar value functions to distributions, a concept first systematically explored by Bellemare et al. (2017). In line with this approach, QR-DQN Dabney et al. (2018b) and IQN Dabney et al. (2018a) have been developed for discrete control tasks. For continuous control, D4PG Barth-Maron et al. (2018), which builds upon DDPG

and incorporates a distributional critic, has achieved state-of-the-art performance. In D4PG, various distributional forms, such as categorical and mixtures of Gaussians, were explored, with the latter minimizing empirical cross-entropy. Shahriari et al. (2022) conducted empirical investigations into the effects of Mahalanobis reweighting and representation learning in a Gaussian critic, concluding that both are beneficial. While our work primarily focuses on the mean value of the value distribution, its analytical and computationally efficient nature makes it applicable to a wide range of settings, distinguishing it from previous methods.

**Policy gradient interpolation** Q-prop Gu et al. (2017b) employs an additional off-policy critic, which serves two purposes: 1) reducing variance in on-policy gradient through control variate; and 2) combining DDPG-style policy updates with on-policy gradient solely on on-policy data. To fully exploit two sources of data, IPG Gu et al. (2017a) interpolates an on-policy gradient with an off-policy gradient Degris et al. (2012), but separately approximates a value function and a state-action value critic, which risks accumulating the compounding error. In practice, it is hard to determine the interpolating parameter. P3O Fakoor et al. (2019) adaptively adjusts the hyperparameter and uses KL divergence to control the off-policyness. PGQL O'Donoghue et al. (2017) combines an entropy-regularized policy gradient with a Q-learning style policy gradient. ACER Wang et al. (2017) enhances sample efficiency through off-policy updates, and the techniques they use, including Retrace Munos et al. (2016), bias-corrected truncated PG with a trust region approach, are complementary to our methods for efficient policy evaluation and stable policy optimization.

**Positive advantage** Though the topic is not widely studied, it serves straightforward purposes – selecting advantageous actions or avoiding bad updates. Tessler et al. (2019) fits an autoregressive actor network to the action that has a positive advantage. van Hasselt (2012) remains the actor unchanged if the TD error is negative to avoid bad updates.

## 9    Conclusion

In this paper, we proposed a novel learning framework that incorporates several key components. Our algorithm is the first to seamlessly integrate on-policy algorithms with off-policy data, striking a balance between stable learning and sample efficiency. We provide theoretical insights and experimental justifications, offering a comprehensive understanding of each algorithmic component. This kind of mixture would be inspiring for future algorithm design.

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

## A  Optimal Baseline

We define gradient components as follows:

$$
\begin{aligned}
g &= u_\theta(s,a)(Q^\pi(s,a) - b(s)) \\
g_1 &= u_\theta(s,a)Q^\pi(s,a) \\
g_2 &= u_\theta(s,a)b(s)
\end{aligned}
\tag{25}
$$

Note that $g = g_1 - g_2$, therefore,

$$
\begin{aligned}
\mathbb{V}[g] &= \mathbb{E}_{\rho_\pi,\pi}[(g - \mathbb{E}_{\rho_\pi,\pi}[g])^T (g - \mathbb{E}_{\rho_\pi,\pi}[g])] \\
&= \mathbb{E}_{\rho_\pi,\pi}\left[ \left((g_1 - \mathbb{E}_{\rho_\pi,\pi}[g_1]) - (g_2 - \mathbb{E}_{\rho_\pi,\pi}[g_2])\right)^T \left((g_1 - \mathbb{E}_{\rho_\pi,\pi}[g_1]) - (g_2 - \mathbb{E}_{\rho_\pi,\pi}[g_2])\right) \right] \\
&= \mathbb{V}[g_1] + \mathbb{V}[g_2] - 2\mathbb{E}_{\rho_\pi,\pi}[(g_1 - \mathbb{E}_{\rho_\pi,\pi}[g_1])^T (g_2 - \overset{0}{\underline{\mathbb{E}_{\rho_\pi,\pi}[g_2]}})] \\
&= \mathbb{V}[g_1] + \mathbb{V}[g_2] - 2\mathbb{E}_{\rho_\pi,\pi}[g_1^T g_2] - 2\overset{0}{(\mathbb{E}_{\rho_\pi,\pi}[g_1])^T \mathbb{E}_{\rho_\pi,\pi}[g_2]} \\
&= \mathbb{V}[g_1] + \mathbb{V}[g_2] - 2\mathbb{E}_{\rho_\pi,\pi}[g_1^T g_2]
\end{aligned}
\tag{26}
$$

Given a state $s$, we can omit the expectation over $\rho_\pi$, which is taken on the state space. It turns out to be:

$$
\begin{aligned}
\mathbb{V}[g|s] &= \mathbb{V}[g_1|s] + \mathbb{V}[g_2|s] - 2\mathbb{E}_\pi[g_1^T g_2|s] \\
&= \mathbb{E}_\pi[u_\theta(s,a)^T u_\theta(s,a)Q^\pi(s,a)^2|s] - \\
&\quad (\mathbb{E}_\pi[u_\theta(s,a)Q^\pi(s,a)|s])^2 + \\
&\quad \mathbb{E}_\pi[u_\theta(s,a)^T u_\theta(s,a)b(s)^2|s] - \\
&\quad 2\mathbb{E}_\pi[u_\theta(s,a)^T u_\theta(s,a)Q^\pi(s,a)b(s)|s]
\end{aligned}
\tag{27}
$$

In an attempt to minimize this variance, using the fact that $\mathbb{V}[g_1|s]$ doesn't dependent on $b(s)$, we can differentiate it w.r.t. $b$, immediately, we get:

$$
b^\star(s) = \frac{\mathbb{E}_\pi[u_\theta(s,a)^T u_\theta(s,a)Q^\pi(s,a)|s]}{\mathbb{E}_\pi[u_\theta(s,a)^T u_\theta(s,a)|s]}
\tag{28}
$$

Define

$$
\tilde{\pi}(a|s) = \frac{\pi(a|s)u_\theta^\top u_\theta}{\mathbb{E}_\pi[u_\theta^\top u_\theta|s]} \qquad l(s,a) = \frac{u_\theta^\top u_\theta}{\mathbb{E}_\pi[u_\theta^\top u_\theta|s]}
\tag{29}
$$

then

$$
b^\star(s) = \mathbb{E}_{\tilde{\pi}}[Q^\pi(s,a)|s]
\tag{30}
$$

**Control Variate**   We are interested in computing $\mathbb{E}_{p(x)}[f(x)]$. However, it may have a high variance. If we introduce another quantity $g(x)$ with a known expectation $\mathbb{E}_{p(x)}[g(x)]$ or whose approximation can be easy. Then we can let $f' = f - a(g - \mathbb{E}[g])$, which will have a same expectation as $\mathbb{E}_{p(x)}[f(x)]$.

We can write out the variance of the new quantity as

$$
\mathbb{V}(f') = \mathbb{V}(f) - 2a\mathrm{cov}(f,g) + a^2\mathbb{V}(g)
\tag{31}
$$

Appropriately adjusting the scalar $a$, we can achieve a lower variance but not change the expectation. It can be analytically solved by minimizing the above quantity w.r.t. $a$

$$
a^\star = \frac{\mathrm{cov}(f,g)}{\mathbb{V}(g)}
\tag{32}
$$

The reduction in ratio can be expressed as

$$
\frac{\mathbb{V}(f')}{\mathbb{V}(f)} = 1 - \mathrm{corr}(f,g)
\tag{33}
$$

The greater correlation between $f$ and $g$ is, the greater variance reduction would attain.

In terms of policy gradient, we can consider $f = u_\theta Q^\pi$ and $g = u_\theta b$, where b is a state-dependent baseline. Since $\mathbb{E}[u_\theta b] = 0$, the new estimator will be $f' = f - a(u_\theta b - 0) = u_\theta(Q^\pi - ab)$, whose optimal value of $a$ is

$$
a^\star = \frac{\mathrm{cov}(u_\theta Q^\pi, u_\theta b)}{\mathbb{V}(u_\theta b)} = \frac{\mathbb{E}[u_\theta^\top u_\theta Q^\pi b]}{\mathbb{E}[u_\theta^\top u_\theta b^2]} = \frac{\mathbb{E}[u_\theta^\top u_\theta Q^\pi]}{\mathbb{E}[u_\theta^\top u_\theta b]} \cdot \frac{1}{b}
\tag{34}
$$

Then $b^\star = a^\star b = \frac{\mathbb{E}[u_\theta^\top u_\theta Q^\pi]}{\mathbb{E}[u_\theta^\top u_\theta b]}$, as what optimal baseline is.

## B   Unified Advantage Estimator

### B.1   Proof of Proposition 3.1

For any $n \in \mathbb{N}^+$, we telescope over residual terms

$$
\sum_{l=0}^{n-1} \gamma^l \delta_{t+l} = \left( \sum_{l=0}^{n-1} \gamma^l r_{t+l} + \gamma^n \Psi_{t+n} - b_t \right) + \sum_{l=1}^{n-1} \gamma^l (\Psi_{t+l} - b_{t+l})
\tag{35}
$$

Since the $\Psi$ is the true quantity we care about, being either $Q^\pi$ or $V^\pi$, thus the Bellman expectation equation is naturally agreed. Denote $z_t = \Psi_t - b_t$, and move the $z_t$ terms from the lefthand side to the righthand size, by taking the expectation of the both sides, it can be shown that

$$
\begin{aligned}
\mathbb{E}_\pi[A_t^{(n)}] &= \mathbb{E}_\pi\left[\sum_{l=0}^{n-1}\gamma^l r_{t+l} + \gamma^n \Psi_{t+n} - b_t\right] \\
&= \Psi_t - b_t \\
&= A_t^{\pi,b}
\end{aligned}
\tag{36}
$$

## B.2 Proof of Equation 5

$$
\begin{aligned}
\hat{A}_t^{\mathrm{UAE}(\gamma,\lambda)} &= (1-\lambda)(\hat{A}_t^{(1)} + \lambda\hat{A}_t^{(2)} + \lambda^2\hat{A}_t^{(3)} + \dots) \\
&= (1-\lambda)\big(\delta_t + \lambda(\delta_t + \gamma\delta_{t+1} - \gamma z_{t+1}) + \lambda^2(\delta_t + \gamma\delta_{t+1} - \gamma z_{t+1} + \gamma^2\delta_{t+2} - \gamma^2 z_{t+2}) + \dots\big) \\
&= (1-\lambda)\Big(\big(\delta_t(1+\lambda+\dots) + \gamma\delta_{t+1}(\lambda+\lambda^2+\dots) + \dots\big) \\
&\qquad - \big(\gamma z_{t+1}(\lambda+\lambda^2+\dots) + \gamma^2 z_{t+2}(\lambda^2+\lambda^3+\dots) + \dots\big)\Big) \\
&= (1-\lambda)\Big(\big(\frac{\delta_t}{1-\lambda} + (\gamma\lambda)\frac{\delta_{t+1}}{1-\lambda} + \dots\big) - \big((\gamma\lambda)\frac{z_{t+1}}{1-\lambda} + (\gamma\lambda)^2\frac{z_{t+2}}{1-\lambda} + \dots\big)\Big) \\
&= \sum_{l=0}^\infty (\gamma\lambda)^l \delta_{t+l} - \sum_{l=1}^\infty (\gamma\lambda)^l z_{t+l} \\
&= \delta_t + \sum_{l=1}^\infty (\gamma\lambda)^l(\delta_{t+l} - z_{t+l})
\end{aligned}
\tag{37}
$$

## B.3 Connection between UAE and SARSA($\lambda$)

We will first show the identity also holds for SARSA($\lambda$), and then connects it with UAE.

$$
\begin{aligned}
G_t^{\lambda,Q} &= (1-\lambda)\sum_{n=1}^\infty G_t^{(n),Q} \\
&= (1-\lambda)(r_t + \gamma Q_{t+1}) + \\
&\qquad (1-\lambda)(r_t + \gamma r_{t+1} + \gamma^2 Q_{t+2}) + \\
&\qquad \dots
\end{aligned}
\tag{38}
$$

By iteratively merging every same reward term of each expansion, we have

$$
\begin{aligned}
G_t^{\lambda,Q} &= r_t + (1-\lambda)\gamma\lambda^0 Q_{t+1} \\
&\qquad + (1-\lambda)\gamma\lambda^1(r_{t+1} + \gamma Q_{t+2}) \\
&\qquad + (1-\lambda)\gamma\lambda^2(r_{t+1} + \gamma r_{t+2} + \gamma^2 Q_{t+3}) \\
&\qquad + \dots \\
&= r_t + \gamma\lambda^0(1-\lambda)Q_{t+1} + \\
&\qquad \gamma\lambda r_{t+2} + \gamma^2\lambda^1(1-\lambda)Q_{t+2} \\
&\qquad \dots
\end{aligned}
\tag{39}
$$

Add $(Q_t - Q_t)$ to Equation 39 without changing the value

$$
\begin{aligned}
G_t^{\lambda,Q} &= Q_t - Q_t + r_t + \gamma Q_{t+1} \\
&\quad - \gamma\lambda Q_{t+1} + \gamma\lambda r_{t+1} + \gamma^2\lambda Q_{t+2} \\
&\quad - \gamma^2\lambda^2 Q_{t+2} + \ldots \\
&= Q_t + \sum_{n=0}^{\infty}(\gamma\lambda)^n \delta_{t+n}^Q
\end{aligned}
\tag{40}
$$

Since we can pull out the first residual term

$$
\begin{aligned}
G_t^{\lambda,Q} &= Q_t + \delta_t^Q + \sum_{n=1}^{\infty}(\gamma\lambda)^n \delta_{t+n}^Q \\
&= Q_t + r_t + \gamma Q_{t+1} - Q_t + \sum_{n=1}^{\infty}(\gamma\lambda)^n \delta_{t+n}^Q \\
&= r_t + \gamma Q_{t+1} + \sum_{n=1}^{\infty}(\gamma\lambda)^n \delta_{t+n}^Q
\end{aligned}
\tag{41}
$$

then it follows that

$$
\begin{aligned}
G_t^{\lambda,Q} - b_t &= r_t + \gamma Q_{t+1} - b_t + \sum_{n=1}^{\infty}(\gamma\lambda)^n \delta_{t+n}^Q \\
&= \delta_t + \sum_{l=1}^{\infty}(\gamma\lambda)^l (\delta_{t+l} - z_{t+l}) \\
&= \hat{A}_t^{\text{UAE}(\gamma,\lambda)}
\end{aligned}
\tag{42}
$$

## B.4   Proof of Theorem 6.4

**Lemma B.1.** *For any $0 \le i < j$, it holds that*

$$
\begin{aligned}
&\text{cov}(r_{t+i} + \gamma V_{t+i+1}^\pi - V_{t+i}^\pi, r_{t+j} + \gamma V_{t+j+1}^\pi - V_{t+j}^\pi) \\
&= \text{cov}(r_{t+i} + \gamma V_{t+i+1}^\pi - V_{t+i}^\pi, r_{t+j} + \gamma Q_{t+j+1}^\pi - Q_{t+j}^\pi)
\end{aligned}
\tag{43}
$$

*Proof.* First note

$$
\begin{aligned}
\mathbb{E}_{\tau|s_t,a_t}[r_{t+j} + \gamma V_{t+j+1}^\pi - V_{t+j}^\pi] &= 0 \\
\mathbb{E}_{\tau|s_t,a_t}[r_{t+j} + \gamma Q_{t+j+1}^\pi - Q_{t+j}^\pi] &= 0
\end{aligned}
\tag{44}
$$

And denote

$$
\begin{aligned}
\delta_i^{V^\pi} &= r_{t+i} + \gamma V_{t+i+1}^\pi - V_{t+i}^\pi \\
\delta_j^{Q^\pi} &= r_{t+j} + \gamma Q_{t+j+1}^\pi - Q_{t+j}^\pi
\end{aligned}
\tag{45}
$$

then

$$
\begin{aligned}
\text{cov}(\delta_i^{V^\pi}, \delta_j^{V^\pi}) &= \mathbb{E}_{\tau|s_t,a_t}[\delta_i^{V^\pi}\delta_j^{V^\pi}] \\
\text{cov}(\delta_i^{V^\pi}, \delta_j^{Q^\pi}) &= \mathbb{E}_{\tau|s_t,a_t}[\delta_i^{V^\pi}\delta_j^{Q^\pi}]
\end{aligned}
\tag{46}
$$

By law of total expectation and Markov property

$$
\begin{aligned}
\operatorname{cov}(\delta_i^{V^\pi}, \delta_j^{Q^\pi}) &= \mathbb{E}_{\tau|s_t,a_t}[\delta_i^{V^\pi}\delta_j^{Q^\pi}] \\
&= \mathbb{E}_{s_{t+i},a_{t+i},s_{t+i+1},s_{t+j},s_{t+j+1}|s_t,a_t}\mathbb{E}_{\tau|s_{t+i},a_{t+i},s_{t+i+1},s_{t+j},s_{t+j+1},s_t,a_t}[\delta_i^{V^\pi}\delta_j^{Q^\pi}] \\
&= \mathbb{E}_{s_{t+i},a_{t+i},s_{t+i+1},s_{t+j},s_{t+j+1}|s_t,a_t}[\delta_i^{V^\pi}\mathbb{E}_{\tau|s_{t+i},a_{t+i},s_{t+i+1},s_{t+j},s_{t+j+1},s_t,a_t}[\delta_j^{Q^\pi}]] \\
&= \mathbb{E}_{s_{t+i},a_{t+i},s_{t+i+1},s_{t+j},s_{t+j+1}|s_t,a_t}[\delta_i^{V^\pi}\mathbb{E}_{a_{t+j+1},a_{t+j}|s_{t+j},s_{t+j+1}}[r_{t+j}+\gamma Q_{t+j+1}^\pi - Q_{t+j}^\pi]] \\
&= \mathbb{E}_{s_{t+i},a_{t+i},s_{t+i+1},s_{t+j},a_{t+j},s_{t+j+1}|s_t,a_t}[\delta_i^{V^\pi}(r_{t+j}+\gamma V_{t+j+1}^\pi - V_{t+j}^\pi)] \\
&= \mathbb{E}_{s_{t+i},a_{t+i},s_{t+i+1},s_{t+j},a_{t+j},s_{t+j+1}|s_t,a_t}[\delta_i^{V^\pi}\delta_j^{V^\pi}] \\
&= \mathbb{E}_{\tau|s_t,a_t}[\delta_i^{V^\pi}\delta_j^{V^\pi}]
\end{aligned}
\tag{47}
$$

$\square$

By law of total variance, we have

$$
\begin{aligned}
\mathbb{V}_{s_t,a_t}[u_\theta A_t^{\mathrm{UAE}(\gamma,\lambda)}] &= \mathbb{E}_{s_t,a_t}[\mathbb{V}_{\tau|s_t,a_t}[u_{\theta_t}A^{\mathrm{UAE}(\gamma,\lambda)}]] + \mathbb{V}_{s_t,a_t}[\mathbb{E}_{\tau|s_t,a_t}[u_\theta A^{\mathrm{UAE}(\gamma,\lambda)}]] \\
&= \mathbb{E}_{s_t,a_t}[u_{\theta_t}^\top u_{\theta_t}\mathbb{V}_{\tau|s_t,a_t}[A^{\mathrm{UAE}(\gamma,\lambda)}]] + \mathbb{V}_{s_t,a_t}[u_\theta \mathbb{E}_{\tau|s_t,a_t}[A^{\mathrm{UAE}(\gamma,\lambda)}]] \\
&= \mathbb{E}_{s_t,a_t}[u_{\theta_t}^\top u_{\theta_t}\mathbb{V}_{\tau|s_t,a_t}[A^{\mathrm{UAE}(\gamma,\lambda)}]] + \mathbb{V}_{s_t,a_t}[u_\theta(Q^\pi - b)]
\end{aligned}
\tag{48}
$$

and similarly for GAE

$$
\mathbb{V}_{s_t,a_t}[u_\theta A_t^{\mathrm{GAE}(\gamma,\lambda)}] = \mathbb{E}_{s_t,a_t}[u_{\theta_t}^\top u_{\theta_t}\mathbb{V}_{\tau|s_t,a_t}[A^{\mathrm{GAE}(\gamma,\lambda)}]] + \mathbb{V}_{s_t,a_t}[u_\theta(Q^\pi - V^\pi)]
\tag{49}
$$

The variance due to sampling a trajectory $\tau$ is

$$
\begin{aligned}
\mathbb{V}_{\tau|s_t,a_t}[A^{\mathrm{UAE}(\gamma,\lambda)}] = \mathbb{V}(\delta_t) &+ \sum_{l=1}^{\infty}(\gamma\lambda)^{2l}\mathbb{V}(\delta_{t+l}-z_{t+l})+ \\
&2\sum_{1\le i<j}(\gamma\lambda)^{i+j}\operatorname{cov}(\delta_{t+i}-z_{t+i},\delta_{t+j}-z_{t+j})+ \\
&2\sum_{j>0}(\gamma\lambda)^j\operatorname{cov}(\delta_t,\delta_{t+j}-z_{t+j})
\end{aligned}
\tag{50}
$$

and

$$
\begin{aligned}
\mathbb{V}_{\tau|s_t,a_t}[A^{\mathrm{GAE}(\gamma,\lambda)}] = \sum_{l=0}^{\infty}(\gamma\lambda)^{2l}\mathbb{V}(\delta_{t+l}^V)+ \\
2\sum_{0\le i<j}(\gamma\lambda)^{i+j}\operatorname{cov}(\delta_{t+i}^V,\delta_{t+j}^V)
\end{aligned}
\tag{51}
$$

We first compare covariance terms.
For $i=0$

$$
\begin{aligned}
\mathrm{UAE}:\ &\operatorname{cov}(r_t+\gamma\Psi_{t+1}-b_t, r_{t+j}+\gamma\Psi_{t+j+1}-\Psi_{t+j}) \\
\mathrm{GAE}:\ &\operatorname{cov}(r_t+\gamma V_{t+1}^\pi-V_t^\pi, r_{t+j}+\gamma V_{t+j+1}^\pi-V_{t+j}^\pi)
\end{aligned}
\tag{52}
$$

the difference of which is

$$
\begin{aligned}
\operatorname{cov}&(r_t+\gamma\Psi_{t+1}-b_t, r_{t+j}+\gamma\Psi_{t+j+1}-\Psi_{t+j}) - \operatorname{cov}(r_t+\gamma V_{t+1}^\pi-V_t^\pi, r_{t+j}+\gamma V_{t+j+1}^\pi-V_{t+j}^\pi) \\
&= \mathbb{E}_{\tau|s_t,a_t}\left[(r_t+\gamma\Psi_{t+1}-b_t)(r_{t+j}+\gamma\Psi_{t+j+1}-\Psi_{t+j})\right] - \mathbb{E}_{\tau|s_t,a_t}\left[(r_t+\gamma V_{t+1}^\pi-V_t^\pi)(r_{t+j}+\gamma V_{t+j+1}^\pi-V_{t+j}^\pi)\right] \\
&\triangleq \chi(0,j|\Psi)
\end{aligned}
\tag{53}
$$

If $\Psi=V^\pi$, then it can be merged as

$$
\begin{aligned}
\chi(0,j|V^\pi) &= \mathbb{E}_{\tau|s_t,a_t}\left[(V_t^\pi-b_t)(r_{t+j}+\gamma V_{t+j+1}^\pi-V_{t+j}^\pi)\right] \\
&= (V_t^\pi-b_t)\mathbb{E}_{\tau|s_t,a_t}\left[r_{t+j}+\gamma V_{t+j+1}^\pi-V_{t+j}^\pi\right] \\
&= 0
\end{aligned}
\tag{54}
$$

If $\Psi = Q^\pi$, by Lemma B.1, and denoting $X_i = (s_{t+i}, a_{t+i})$ we have

$$
\begin{aligned}
\chi(0,j|Q^\pi) &= \mathbb{E}_{\tau|X_0} \left[ \left( \gamma(Q_{t+1}^\pi - V_{t+1}^\pi) - (b_t - V_t^\pi) \right) (r_{t+j} + \gamma Q_{t+j+1}^\pi - Q_{t+j}^\pi) \right] \\
&= \mathbb{E}_{X_1, X_j, X_{j+1}|X_0} \left[ \left( \gamma(Q_{t+1}^\pi - V_{t+1}^\pi) - (b_t - V_t^\pi) \right) (r_{t+j} + \gamma Q_{t+j+1}^\pi - Q_{t+j}^\pi) \right] \\
&= \mathbb{E}_{X_1, X_j|X_0} \mathbb{E}_{X_{j+1}|X_1, X_j, X_0} \left[ \left( \gamma(Q_{t+1}^\pi - V_{t+1}^\pi) - (b_t - V_t^\pi) \right) (r_{t+j} + \gamma Q_{t+j+1}^\pi - Q_{t+j}^\pi) \right] \\
&= \mathbb{E}_{X_1, X_j|X_0} \left[ \left( \gamma(Q_{t+1}^\pi - V_{t+1}^\pi) - (b_t - V_t^\pi) \right) \mathbb{E}_{X_{j+1}|X_1, X_j, X_0} \left[ (r_{t+j} + \gamma Q_{t+j+1}^\pi - Q_{t+j}^\pi) \right] \right] \\
&= \mathbb{E}_{X_1, X_j|X_0} \left[ \left( \gamma(Q_{t+1}^\pi - V_{t+1}^\pi) - (b_t - V_t^\pi) \right) 0 \right] \\
&= 0
\end{aligned}
\tag{55}
$$

For $i > 0$

$$
\begin{aligned}
&\text{UAE}: \ \text{cov}(r_{t+i} + \gamma \Psi_{t+i+1} - \Psi_{t+i}, r_{t+j} + \gamma \Psi_{t+j+1} - \Psi_{t+j}) \\
&\text{GAE}: \ \text{cov}(r_{t+i} + \gamma V_{t+i+1}^\pi - V_{t+i}^\pi, r_{t+j} + \gamma V_{t+j+1}^\pi - V_{t+j}^\pi)
\end{aligned}
\tag{56}
$$

If $\Psi = V^\pi$, it is obvious that $\chi(i,j|V^\pi) = 0$. If $\Psi = Q^\pi$, similarly we have

$$
\begin{aligned}
\chi(i,j|Q^\pi) &= \mathbb{E}_{\tau|X_0} \left[ \left( \gamma(Q_{t+i+1}^\pi - V_{t+i+1}^\pi) - (Q_{t+i}^\pi - V_{t+i}^\pi) \right) (r_{t+j} + \gamma Q_{t+j+1}^\pi - Q_{t+j}^\pi) \right] \\
&= \mathbb{E}_{X_{i+1}, X_j, X_{j+1}|X_0} \left[ \left( \gamma(Q_{t+i+1}^\pi - V_{t+i+1}^\pi) - (Q_{t+i}^\pi - V_{t+i}^\pi) \right) (r_{t+j} + \gamma Q_{t+j+1}^\pi - Q_{t+j}^\pi) \right] \\
&= \mathbb{E}_{X_{i+1}, X_j|X_0} \mathbb{E}_{X_{j+1}|X_{i+1}, X_j, X_0} \left[ \left( \gamma(Q_{t+i+1}^\pi - V_{t+i+1}^\pi) - (Q_{t+i}^\pi - V_{t+i}^\pi) \right) (r_{t+j} + \gamma Q_{t+j+1}^\pi - Q_{t+j}^\pi) \right] \\
&= \mathbb{E}_{X_{i+1}, X_j|X_0} \left[ \left( \gamma(Q_{t+i+1}^\pi - V_{t+i+1}^\pi) - (Q_{t+i}^\pi - V_{t+i}^\pi) \right) \mathbb{E}_{X_{j+1}|X_{i+1}, X_j, X_0} \left[ (r_{t+j} + \gamma Q_{t+j+1}^\pi - Q_{t+j}^\pi) \right] \right] \\
&= \mathbb{E}_{X_{i+1}, X_j|X_0} \left[ \left( \gamma(Q_{t+i+1}^\pi - V_{t+i+1}^\pi) - (Q_{t+i}^\pi - V_{t+i}^\pi) \right) 0 \right] \\
&= 0
\end{aligned}
$$
$$\tag{57}$$

And next we will focus on the variance terms.

For $l = 0$

$$
\begin{aligned}
&\mathbb{V}(\delta_t) - \mathbb{V}(\delta_t^{V^\pi}) \\
&= \mathbb{E}_{\tau|s_t, a_t}[(r_t + \gamma \Psi_{t+1} - Q_t^\pi)^2] - \mathbb{E}_{\tau|s_t, a_t}[(r_t + \gamma V_{t+1}^\pi - Q_t^\pi)^2] \\
&= \mathbb{E}_{\tau|s_t, a_t}[(r_t + \gamma \Psi_{t+1})^2] - (Q_t^\pi)^2 - (\mathbb{E}_{\tau|s_t, a_t}[(r_t + \gamma V_{t+1}^\pi)^2] - (Q_t^\pi)^2) \\
&= \mathbb{E}_{\tau|s_t, a_t}[(r_t + \gamma \Psi_{t+1})^2] - \mathbb{E}_{\tau|s_t, a_t}[(r_t + \gamma V_{t+1}^\pi)^2] \\
&= \mathbb{E}_{\tau|s_t, a_t}[\gamma^2(\Psi_{t+1}^2 - (V_{t+1}^\pi)^2) + 2\gamma(\Psi_{t+1} - V_{t+1}^\pi)] \\
&= \mathbb{E}_{\tau|s_t, a_t}[\gamma^2(\Psi_{t+1}^2 - (V_{t+1}^\pi)^2)] + 2\gamma \mathbb{E}_{\tau|s_t, a_t}[\Psi_{t+1} - V_{t+1}^\pi] \\
&= \gamma^2 \mathbb{E}_{\tau|s_t, a_t}[(\Psi_{t+1}^2 - (V_{t+1}^\pi)^2)] \\
&= \gamma^2 \mathbb{E}_{s_{t+1}, a_{t+1}|s_t, a_t}[\Psi_{t+1}^2] - \mathbb{E}_{s_{t+1}|s_t, a_t}[(\mathbb{E}_{a_{t+1}|s_{t+1}, s_t, a_t}[Q_{t+1}^\pi])^2] \\
&= \begin{cases} \gamma^2 \mathbb{E}_{s_{t+1}|s_t, a_t}[\mathbb{V}_{a_{t+1}|s_{t+1}, s_t, a_t}(Q_{t+1}^\pi)] & \text{if } \Psi = Q^\pi \\ 0 & \text{if } \Psi = V^\pi \end{cases} \\
&= \gamma^2 \mathbb{E}_{\tau|s_t, a_t}[(\Psi_{t+1} - V_{t+1}^\pi)^2]
\end{aligned}
\tag{58}
$$

For $l > 0$

$$
\begin{aligned}
&\mathbb{V}(\delta_{t+l}) - \mathbb{V}(\delta_{t+l}^{V^\pi}) \\
&= \mathbb{E}_{\tau|s_t, a_t}[(r_{t+l} + \gamma \Psi_{t+l+1} - \Psi_{t+l})^2] - \mathbb{E}_{\tau|s_t, a_t}[(r_{t+l} + \gamma V_{t+l+1}^\pi - V_{t+l}^\pi)^2] \\
&= \mathbb{E}_{\tau|s_t, a_t}[(r_{t+l} + \gamma \Psi_{t+l+1})^2] - \mathbb{E}_{\tau|s_t, a_t}[(r_{t+l} + \gamma V_{t+l+1}^\pi)^2] - \mathbb{E}_{s_{t+l}, a_{t+l}|s_t, a_t}[(\Psi_{t+l} - V_{t+l}^\pi)^2] \\
&= \mathbb{E}_{\tau|s_t, a_t}[\gamma^2(\Psi_{t+l+1}^2 - (V_{t+l+1}^\pi)^2) + 2\gamma r_{t+l}(\Psi_{t+l+1} - V_{t+l+1}^\pi)] - \mathbb{E}_{s_{t+l}, a_{t+l}|s_t, a_t}[(\Psi_{t+l} - V_{t+l}^\pi)^2] \\
&= \mathbb{E}_{\tau|s_t, a_t}[\gamma^2(\Psi_{t+l+1}^2 - (V_{t+l+1}^\pi)^2)] + 2\gamma r_{t+l} \mathbb{E}_{\tau|s_t, a_t}[\Psi_{t+l+1} - V_{t+l+1}^\pi] - \mathbb{E}_{s_{t+l}, a_{t+l}|s_t, a_t}[(\Psi_{t+l} - V_{t+l}^\pi)^2] \\
&= \gamma^2 \mathbb{E}_{\tau|s_t, a_t}[(\Psi_{t+l+1}^2 - (V_{t+l+1}^\pi)^2)] - \mathbb{E}_{s_{t+l}, a_{t+l}|s_t, a_t}[(\Psi_{t+l} - V_{t+l}^\pi)^2] \\
&= \gamma^2 \mathbb{E}_{s_{t+l+1}, a_{t+l+1}|s_t, a_t}[\Psi_{t+l+1}^2] - \mathbb{E}_{s_{t+l+1}|s_t, a_t}[(\mathbb{E}_{a_{t+l+1}|s_{t+l+1}, s_t, a_t}[Q_{t+l+1}^\pi])^2] - \mathbb{E}_{s_{t+l}, a_{t+l}|s_t, a_t}[(\Psi_{t+l} - V_{t+l}^\pi)^2] \\
&= \gamma^2 \mathbb{E}_{\tau|s_t, a_t}[(\Psi_{t+l+1} - V_{t+l+1}^\pi)^2] - \mathbb{E}_{\tau|s_t, a_t}[(\Psi_{t+l} - V_{t+l}^\pi)^2]
\end{aligned}
$$
$$\tag{59}$$

Combining all the results above, we have

$$
\begin{aligned}
\mathbb{V}_{s_t,a_t}&[u_\theta A_t^{\mathrm{UAE}(\gamma,\lambda)}] - \mathbb{V}_{s_t,a_t}[u_\theta A_t^{\mathrm{GAE}(\gamma,\lambda)}] \\
&= \mathbb{E}_{s_t,a_t}\left[u_\theta^\top u_\theta\left(\sum_{l=0}^{\infty}(\gamma\lambda)^{2l}\cdot\gamma^2(1-\lambda^2)\mathbb{E}_{s_{t+l+1},a_{t+l+1}}\left[(\Psi - V^\pi)^2\right]\right)\right] + \\
&\quad \mathbb{E}_{s_t,a_t}[u_\theta^\top u_\theta(b^2 - V^{\pi 2} - 2Q^\pi(b - V^\pi))]
\end{aligned}
\tag{60}
$$

## B.5 Proof of corollary 6.5

If $\Psi = V^\pi$, then the first term of Equation 60 vanishes, it reduces to the difference between the variance of policy gradient w.r.t. different baselines. Since $b$ reduces variance no less than $V^\pi$, it follows that

$$
\begin{aligned}
\mathbb{V}_{s_t,a_t}[u_\theta A_t^{\mathrm{UAE}(\gamma,\lambda)}] - \mathbb{V}_{s_t,a_t}[u_\theta A_t^{\mathrm{GAE}(\gamma,\lambda)}] &= \mathbb{E}_{s_t,a_t}[u_\theta^\top u_\theta(b^2 - V^{\pi 2} - 2Q^\pi(b - V^\pi))] \\
&= \mathbb{E}_{s_t,a_t}[u_\theta^\top u_\theta(Q^\pi - b)^2] - \mathbb{E}_{s_t,a_t}[u_\theta^\top u_\theta(Q^\pi - V^\pi)^2] \\
&\leq 0
\end{aligned}
\tag{61}
$$

# C Residual Baseline

## C.1 Proof of Theorem 6.6

Our analysis relies on the fact that for a sufficient large segment length $T$, the empirical distribution $p_\mathcal{D}$ is an approximation of a mixture of the joint distributions of the past policy sequences $d_\beta(s,a) = \frac{1}{k}\sum_{i=0}^{k-1} d_{\pi_{n-i}}(s,a)$, whose marginal state distribion is $d_\beta(s) = \int_\mathcal{A} d_\beta(s,a)da$, and conditional action distribution $\beta(a|s) = \frac{d_\beta(s,a)}{d_\beta(s)}$. For any past policy, the joint distribution and marginal state distribution are abbreviated as $d_{n-i}(s,a)$ and $d_{n-i}(s)$ respectively for simplicity's sake. And $\pi_n$ is recurrently referenced as $\pi$ whenever noticed.

Since $\sup_i D_{\mathrm{TV}}(\frac{d_{n-i}(s,a)}{d_\beta(s)}||\pi) < \frac{\epsilon}{4}$ and $\mathbb{E}_\pi[|\mathrm{r}_{\phi^\star}(s,a)|] < \frac{\epsilon}{2}$ for any $s \in \mathcal{S}$, it follows that

$$
\begin{aligned}
\left|\sum_{i=0}^{k-1}\int_\mathcal{A}\left(\frac{d_{n-i}(s)}{d_\beta(s)}\pi_{n-i} - (1+\mathrm{r}_{\phi^\star})\pi_n\right)Q_w da\right| &= \left|\sum_{i=0}^{k-1}\int_\mathcal{A}\left(\frac{d_{n-i}(s,a)}{d_\beta(s)} - (1+\mathrm{r}_{\phi^\star})\pi_n\right)Q_w da\right| \\
&\leq \sum_{i=0}^{k-1}\left|\int_\mathcal{A}\left(\frac{d_{n-i}(s,a)}{d_\beta(s)} - (1+\mathrm{r}_{\phi^\star})\pi_n\right)Q_w da\right| \\
&\leq \sum_{i=0}^{k-1}\int_\mathcal{A}\left|\frac{d_{n-i}(s,a)}{d_\beta(s)} - (1+\mathrm{r}_{\phi^\star})\pi_n\right|\left|Q_w\right|da \\
&\leq M\sum_{i=0}^{k-1}\int_\mathcal{A}\left|\frac{d_{n-i}(s,a)}{d_\beta(s)} - (1+\mathrm{r}_{\phi^\star})\pi_n\right|da \\
&\leq M\sum_{i=0}^{k-1}\left(2\cdot\left(\frac{1}{2}\int_\mathcal{A}\left|\frac{d_{n-i}(s,a)}{d_\beta(s)} - \pi_n\right|da\right) + \int_\mathcal{A}\pi_n|\mathrm{r}_{\phi^\star}|da\right) \\
&< M\sum_{i=0}^{k-1}\left(2\frac{\epsilon}{4} + \frac{\epsilon}{2}\right) \\
&= Mk\epsilon
\end{aligned}
\tag{62}
$$

Henceforth

$$
\begin{aligned}
&\left| \mathbb{E}_\beta[Q_w] - \mathbb{E}_\pi[(1 + \mathrm{r}_{\phi^\star})Q_w] \right| \\
&= \left| \frac{1}{k} \sum_{i=1}^{k-1} \left( \mathbb{E}_{\pi_{n-i}} \left[ \frac{d_{n-i}(s)}{d_\beta(s)} Q_w \right] + \mathbb{E}_\pi \left[ \frac{d_\pi(s)}{d_\beta(s)} Q_w \right] \right) - \mathbb{E}_\pi[(1 + \mathrm{r}_{\phi^\star})Q_w] \right| \\
&= \left| \frac{1}{k} \sum_{i=1}^{k-1} \left( \mathbb{E}_{\pi_{n-i}} \left[ \frac{d_{n-i}(s)}{d_\beta(s)} Q_w \right] - \mathbb{E}_\pi \left[ \left( k + k\mathrm{r}_{\phi^\star} - \frac{d_\pi(s)}{d_\beta(s)} \right) Q_w \right] \right) \right| \\
&\overset{(a)}{=} \left| \frac{1}{k} \sum_{i=1}^{k-1} \left( \mathbb{E}_{\pi_{n-i}} \left[ \frac{d_{n-i}(s)}{d_\beta(s)} Q_w \right] - \mathbb{E}_\pi \left[ \left( (k-1)\frac{d_\pi(s)}{d_\beta(s)} + k(1 - \frac{d_\pi(s)}{d_\beta(s)}) + k\mathrm{r}_{\phi^\star} \right) Q_w \right] \right) \right| \\
&= \left| \frac{1}{k} \sum_{i=1}^{k-1} \int_\mathcal{A} \left( \frac{d_{n-i}(s)}{d_\beta(s)} \pi_{n-i} - \frac{d_\pi(s)}{d_\beta(s)} \pi \right) Q_w da - k\mathbb{E}_\pi \left[ (1 - \frac{d_\pi(s)}{d_\beta(s)} + \mathrm{r}_{\phi^\star})Q_w \right] \right| \\
&\overset{(b)}{=} \left| \frac{1}{k} \sum_{i=0}^{k-1} \int_\mathcal{A} \left( \frac{d_{n-i}(s)}{d_\beta(s)} \pi_{n-i} - \frac{d_\pi(s)}{d_\beta(s)} \pi \right) Q_w da - k\mathbb{E}_\pi \left[ (1 - \frac{d_\pi(s)}{d_\beta(s)} + \mathrm{r}_{\phi^\star})Q_w \right] \right| \\
&= \left| \frac{1}{k} \sum_{i=0}^{k-1} \int_\mathcal{A} \left( \frac{d_{n-i}(s)}{d_\beta(s)} \pi_{n-i} - \frac{d_\pi(s)}{d_\beta(s)} \pi - (1 - \frac{d_\pi(s)}{d_\beta(s)} + \mathrm{r}_{\phi^\star})\pi \right) Q_w da \right| \\
&= \left| \frac{1}{k} \sum_{i=0}^{k-1} \int_\mathcal{A} \left( \frac{d_{n-i}(s)}{d_\beta(s)} \pi_{n-i} - (1 + \mathrm{r}_{\phi^\star})\pi \right) Q_w da \right| \\
&= \frac{1}{k} \left| \sum_{i=0}^{k-1} \int_\mathcal{A} \left( \frac{d_{n-i}(s)}{d_\beta(s)} \pi_{n-i} - (1 + \mathrm{r}_{\phi^\star})\pi_n \right) Q_w da \right| \\
&\leq \frac{1}{k} M k \epsilon \\
&= M \epsilon
\end{aligned}
\tag{63}
$$

where (a) holds by adding $k\frac{d_\pi(s)}{d_\beta(s)} - k\frac{d_\pi(s)}{d_\beta(s)}$ without changing the quantity, and (b) by noting $\frac{d_{n-i}(s)}{d_\beta(s)}\pi_{n-i} - \frac{d_\pi(s)}{d_\beta(s)}\pi = 0$ when $i = 0$. Let $g_{\mathrm{r}_{\phi^\star}}(s) = \mathbb{E}_\pi[\nabla_\phi \mathrm{r}_{\phi^\star} Q_w]$, by assumption it follows that

$$
\|g_{\mathrm{r}_{\phi^\star}}(s)\| \leq GM
\tag{64}
$$

Then

$$
\begin{aligned}
\|\nabla_\phi \mathcal{J}(\phi^\star)\| &= \|\mathbb{E}_{d_\beta}\{2 \cdot (\mathbb{E}_\pi[(1 + \mathrm{r}_{\phi^\star})Q_w] - Q_w)g_{\mathrm{r}_{\phi^\star}}(s)\}\| \\
&\leq \mathbb{E}_{(s,a)\sim d_\beta(s,a)}\{2 \cdot |(\mathbb{E}_\pi[(1 + \mathrm{r}_{\phi^\star})Q_w] - Q_w)|\|g_{\mathrm{r}_{\phi^\star}}(s)\|\} \\
&\leq \mathbb{E}_{s\sim d_\beta(s)}\{2 \cdot |\mathbb{E}_\pi[(1 + \mathrm{r}_{\phi^\star})Q_w] - \mathbb{E}_{a\sim\beta}[Q_w]|\|g_{\mathrm{r}_{\phi^\star}}(s)\|\} \\
&\leq GM^2\epsilon
\end{aligned}
\tag{65}
$$

## C.2 Proof of Corollary 6.7

By $D_{\mathrm{TV}}^2(p||q) \leq \frac{1}{2} D_{\mathrm{KL}}(p||q)$ (Pinsker's inequality), we have $\sup_s D_{\mathrm{TV}}(\tilde{\pi}||\pi) < \frac{\sqrt{\epsilon}}{2}$

$$
\begin{aligned}
\mathbb{E}_{d_\beta}\big[(b_{\phi^\star}^{\tilde{\pi}} - b_{\phi^\star}^{\pi})^2\big] &= \mathbb{E}_{d_\beta}\Big[\Big(\int_{\mathcal{A}} (\tilde{\pi} - \pi)(1 + \mathrm{r}_{\phi^\star})Q_w da\Big)^2\Big] \\
&= \mathbb{E}_{d_\beta}\Big[\big|\int_{\mathcal{A}} (\tilde{\pi} - \pi)(1 + \mathrm{r}_{\phi^\star})Q_w da\big|^2\Big] \\
&\leq \mathbb{E}_{d_\beta}\Big[\Big(\int_{\mathcal{A}} |\tilde{\pi} - \pi|(1 + |\mathrm{r}_{\phi^\star}|)|Q_w|da\Big)^2\Big] \\
&< ((K+1)M)^2 \mathbb{E}_{d_\beta}\Big[\big(2 \cdot \frac{1}{2}\int_{\mathcal{A}} |\tilde{\pi} - \pi|da\big)^2\Big] \\
&< ((K+1)M)^2 \mathbb{E}_{d_\beta}\Big[\big(2 \cdot \frac{\sqrt{\epsilon}}{2}\big)^2\Big] \\
&= ((K+1)M)^2 \epsilon
\end{aligned}
\tag{66}
$$

And with the fact that $(a+b)^2 \leq 2(a^2 + b^2)$, we have that

$$
\begin{aligned}
\mathbb{E}_{d_\beta}[(Q_w - b_{\phi^\star}^{\tilde{\pi}})^2] &= \mathbb{E}_{d_\beta}[(Q_w - b_{\phi^\star}^{\pi} + b_{\phi^\star}^{\pi} - b_{\phi^\star}^{\tilde{\pi}})^2] \\
&\leq 2 \cdot \Big(\mathbb{E}_{d_\beta(s,a)}[(Q_w - b_{\phi^\star}^{\pi})^2] + \mathbb{E}_{d_\beta(s)}\big[(b_{\phi^\star}^{\tilde{\pi}} - b_{\phi^\star}^{\pi})^2\big]\Big) \\
&< 2\mathcal{J}(\phi^\star) + 2((K+1)M)^2 \epsilon
\end{aligned}
\tag{67}
$$

# D Practical Algorithm

## D.1 Contraction properties in mean and variance

$$
\begin{aligned}
\|\mathbb{E}\mathcal{T}^\pi Z_1 - \mathbb{E}\mathcal{T}^\pi Z_2\|_\infty &= \gamma\|\mathbb{E}_{s',a'}[\mathbb{E}Z_1 - \mathbb{E}Z_2]\|_\infty \\
&\leq \gamma\|\mathbb{E}Z_1 - \mathbb{E}Z_2\|_\infty
\end{aligned}
\tag{68}
$$

$$
\begin{aligned}
\|\mathbb{V}\mathcal{T}^\pi Z_1 - \mathbb{V}\mathcal{T}^\pi Z_2\|_\infty &= \sup_{s,a} |\mathbb{V}\mathcal{T}^\pi Z_1(s,a) - \mathbb{V}\mathcal{T}^\pi Z_2(s,a)| \\
&= \sup_{s,a} |\mathbb{E}[\mathbb{V}\mathcal{T}^\pi Z_1(s,a) - \mathbb{V}\mathcal{T}^\pi Z_2(s,a)]| \\
&= \sup_{s,a} \gamma^2 |\mathbb{E}[\mathbb{V}Z_1(S',A') - \mathbb{V}Z_2(S',A')]| \\
&\leq \sup_{s',a'} \gamma^2 |\mathbb{V}Z_1(s',a') - \mathbb{V}Z_2(s',a')| \\
&= \gamma^2 \|\mathbb{V}Z_1 - \mathbb{V}Z_2\|_\infty
\end{aligned}
\tag{69}
$$

## D.2 Equivalence between KL divergence and MSE

Suppose $(\mu_i, \sigma_i), i = 1, 2$ indexed by 1 is the parameters of the target distribution, and 2 of the learnable distribution. Since $\sigma_1$ and $\sigma_2$ are constantly equal as $\sigma$, it means that both of them are not parameterized, then it immediately follows with the definition of the KL divergence between two normal distributions

$$
\begin{aligned}
D_{\mathrm{KL}}(\mathcal{N}(\mu_1, \sigma_1), \mathcal{N}(\mu_2(\phi), \sigma_2)) &= \log\Big(\frac{\sigma_2}{\sigma_1}\Big) + \frac{\sigma_1^2 + (\mu_1 - \mu_2(\phi))^2}{2\sigma_2^2} - \frac{1}{2} \\
&= \log\Big(\frac{\sigma}{\sigma}\Big) + \frac{\sigma^2}{2\sigma^2} - \frac{1}{2} + \frac{(\mu_1 - \mu_2(\phi))^2}{2\sigma^2} \\
&= \frac{1}{\sigma^2}\mathrm{MSE}(\mu_1, \mu_2(\phi))
\end{aligned}
\tag{70}
$$

### D.3 Distributionl critic update

Suppose the learnable distribution is parameterized as $\mathcal{N}(\mu(\phi), \sigma(\zeta))$. Taking gradient of the KL divergence loss, it can be attained

$$
\nabla_{(\phi,\sigma)} D_{\mathrm{KL}}(\mathcal{N}(\mu_1, \sigma_1), \mathcal{N}(\mu_2(\phi), \sigma_2(\zeta))) = -\frac{\left((\sigma_1^2 - \sigma_2^2(\zeta)) + (\mu_1 - \mu_2(\phi))^2\right)\nabla_\zeta \sigma_2(\zeta)}{\sigma_2^3(\zeta)} - \frac{(\mu_1 - \mu_2(\phi))\nabla_\phi \mu_2(\phi)}{\sigma_2^2(\zeta)}
$$
$$
= \Delta\sigma_2 + \Delta\mu_2
$$
(71)

In practice, the sampling is involved as transition dynamics evolves, thus the $\mu_1 = r + \gamma \cdot \mathrm{mean}(Z_{\bar{w}}(s', a'))$, and $\sigma_1 = \gamma \cdot \mathrm{stddev}(Z_{\bar{w}}(s', a'))$. And $\mu_2 = \mathrm{mean}(Z_w(s, a))$, and $\sigma_2 = \mathrm{stddev}(Z_w(s, a))$. It is clear that the parameters of the learnable distribution is always chasing for discounted ones of the target distribution.

### D.4 Connection between DPO and SAC

Our off-policy gradient can be viewed as an optimistic likelihood ratio gradient estimator of SAC, that is

$$
\nabla_\theta \mathcal{J}_{\text{off-policy}}(\theta) = \nabla_\theta \mathbb{E}_{s\sim\mathcal{D}}\left[-D_{\mathrm{KL}}(\pi_\theta(\cdot|s)\,\Big\|\,\frac{\frac{1}{\alpha}\exp A^+(s,\cdot)}{N(s)})\right] \triangleq \nabla_\theta \mathcal{J}_{\text{OptSAC}}(\theta)
$$
(72)

where $N(s)$ is the partition function.

Since that

$$
\mathbb{E}_{s\sim D, a\sim\pi_\theta}[\nabla_\theta \log \pi_\theta(a|s) \log N(s)] = \int d_\mathcal{D}(s) \int \nabla_\theta \pi_\theta(a|s) \log N(s) da ds
$$
$$
= \int d_\mathcal{D}(s) \log N(s)(\nabla_\theta \int \pi_\theta(a|s) da) ds
$$
$$
= 0
$$
(73)

and

$$
\mathbb{E}_{s\sim D, a\sim\pi_\theta}[\nabla_\theta \log \pi_\theta(a|s)] = \int d_\mathcal{D}(s) \int \pi_\theta(a|s) \frac{\nabla_\theta \pi_\theta(a|s)}{\pi_\theta(a|s)} da ds
$$
$$
= \int d_\mathcal{D}(s) \int \nabla_\theta \pi_\theta(a|s) da ds
$$
$$
= \int d_\mathcal{D}(s)(\nabla_\theta \int \pi_\theta(a|s) da) ds
$$
$$
= 0
$$
(74)

Then

$$
\nabla_\theta \mathcal{J}_{\text{OptSAC}}(\theta) = \nabla_\theta \mathbb{E}_{s\sim D, a\sim\pi_\theta}[\log \pi_\theta(a|s)(A^+(s,a) - \alpha \log \pi_\theta(a|s) - \log N(s))]
$$
$$
= \mathbb{E}_{s\sim D, a\sim\pi_\theta}[\nabla_\theta \log \pi_\theta(a|s)(A^+(s,a)
$$
$$
- \alpha \log \pi_\theta(a|s) - \log N(s))] - \mathbb{E}_{s\sim D, a\sim\pi_\theta}[\nabla_\theta \log \pi_\theta(a|s) \log N(s)]
$$
$$
+ \mathbb{E}_{s\sim D, a\sim\pi_\theta}[\nabla_\theta \log \pi_\theta(a|s)]
$$
$$
= \mathbb{E}_{s\sim D, a\sim\pi_\theta}[\nabla_\theta \log \pi_\theta(a|s)(A^+(s,a) - \alpha \log \pi_\theta(a|s))]
$$
$$
= \nabla_\theta \mathcal{J}_{\text{off-policy}}(\theta)
$$
(75)

### D.5 Proof of Theorem 6.8

Since

$$
|\pi_k \mathrm{r}_{\phi_k}| \le K
$$
(76)

and

$$
\lim_{k\to\infty} \pi_k \mathrm{r}_{\phi_k} = \pi^\star \mathrm{r}_{\phi^\star}
$$
(77)

by Lebesgue bounded convergence theorem, we have

$$\lim_{k\to\infty} \mathbb{E}_{\pi_k}[\mathrm{r}_{\phi_k}] = \mathbb{E}_{\pi^\star}[\mathrm{r}_{\phi^\star}] \tag{78}$$

Then

$$
\begin{aligned}
&\int_{\mathcal{A}} \pi^\star(a|s)\mathrm{r}_{\phi^\star}(s,a)Q^{\pi^\star}(s,a)da \\
&= \int_{\mathcal{A}\setminus\mathcal{A}^0} \pi^\star(a|s)\mathrm{r}_{\phi^\star}(s,a)Q^{\pi^\star}(s,a)da \\
&= Q^\star \int_{\mathcal{A}\setminus\mathcal{A}^0} \pi^\star(a|s)\mathrm{r}_{\phi^\star}(s,a)da \\
&= Q^\star \int_{\mathcal{A}} \pi^\star(a|s)\mathrm{r}_{\phi^\star}(s,a)da \\
&= Q^\star \mathbb{E}_{\pi^\star}[\mathrm{r}_{\phi^\star}] \\
&= 0
\end{aligned}
\tag{79}
$$

where $Q^\star$ stands for the maximum action-value. As the $\pi^\star$ is deterministic, then $Q^{\pi^\star}(s,a) = Q^\star$ for any $a \in \mathcal{A}\setminus\mathcal{A}^0$. Thus it follows

$$
\begin{aligned}
\limsup_{k\to\infty} A_k^+ &= \limsup_{k\to\infty}(Q^{\pi_k}(s,a) - b_{\phi_k}(s))^+ \\
&= \limsup_{k\to\infty}(Q^{\pi_k}(s,a) - b_{\phi_k}(s))^+ \\
&= (Q^{\pi^\star}(s,a) - \mathbb{E}[(1+\mathrm{r}_{\phi^\star}(s,a))Q^{\pi^\star}(s,a)])^+ \\
&= (Q^{\pi^\star}(s,a) - V^{\pi^\star}(s) - Q^\star \int_{\mathcal{A}\setminus\mathcal{A}^0} \pi^\star(a|s)\mathrm{r}_{\phi^\star}(s,a)da)^+ \\
&= (Q^{\pi^\star}(s,a) - V^{\pi^\star}(s))^+ = 0
\end{aligned}
\tag{80}
$$

Last equality holds for that $V^{\pi^\star}(s) = \max_{a\in\mathcal{A}} Q^{\pi^\star}(s,a)$.

### D.6 Proof of Theorem 6.9

**Lemma D.1.** *Gu et al. (2017a)*

$$\|\rho^\pi - \rho^\beta\|_1 \le 2tD_{TV}^{max}(\pi||\beta) \le 2t\sqrt{D_{KL}^{max}(\pi||\beta)} \tag{81}$$

which can be adopted from Ross et al. (2011) Kahn et al. (2017) Schulman et al. (2015) Janner et al. (2019).

Denote

$$\bar{L}_\pi(\tilde\pi) = \eta(\pi) + \mathbb{E}_{\rho^\pi,\tilde\pi}[Q^\pi - b_\phi^\pi] \tag{82}$$

then

$$
\begin{aligned}
|\eta(\tilde\pi) - \bar{L}_\pi(\tilde\pi)| &= |\mathbb{E}_{\rho^{\tilde\pi},\tilde\pi}[Q^\pi - b_\phi^\pi] - \mathbb{E}_{\rho^\pi,\tilde\pi}[Q^\pi - b_\phi^\pi]| \\
&= \sum_{t=0}^{\infty}\gamma^t|\mathbb{E}_{\rho_t^{\tilde\pi}}\left[\mathbb{E}_{\tilde\pi}[Q^\pi - b_\phi^\pi]\right] - \mathbb{E}_{\rho_t^\pi}\left[\mathbb{E}_{\tilde\pi}[Q^\pi - b_\phi^\pi]\right]| \\
&\le \Upsilon\sum_{t=0}^{\infty}\gamma^t\|\rho_t^{\tilde\pi} - \rho_t^\pi\|_1 \\
&\le 2\Upsilon\sum_{t=0}^{\infty}\gamma^t t\sqrt{D_{KL}^{max}(\pi||\tilde\pi)} \\
&= \frac{2\gamma\Upsilon}{(1-\gamma)^2}\sqrt{D_{KL}^{max}(\pi||\tilde\pi)}
\end{aligned}
\tag{83}
$$

And we relate $L_\pi(\tilde\pi)$ to $\bar{L}_\pi(\tilde\pi)$

$$|\bar{L}_\pi(\tilde\pi) - L_\pi(\tilde\pi)| = |\eta(\pi) + \mathbb{E}_{\rho^\pi,\tilde\pi}[Q^\pi - b_\phi^\pi] - \eta(\pi) - \omega\mathbb{E}_{\rho^\pi,\tilde\pi}[Q^\pi - b_\phi^\pi] - (1-\omega)\mathbb{E}_{\rho^\beta,\tilde\pi}[(Q_w - b_\phi^{\tilde\pi})^+ + \alpha\log\tilde\pi]|$$

$$= (1-\omega)|\mathbb{E}_{\rho^\pi,\tilde\pi}[Q^\pi - b_\phi^\pi] - \mathbb{E}_{\rho^\beta,\tilde\pi}[(Q_w - b_\phi^{\tilde\pi})^+ + \alpha\log\tilde\pi]|$$

$$\leq (1-\omega)\Bigg(\underbrace{|\mathbb{E}_{\rho^\pi,\tilde\pi}[Q^\pi - b_\phi^\pi] - \mathbb{E}_{\rho^\pi,\tilde\pi}[Q_w - b_\phi^{\tilde\pi}]|}_{(1)} +$$

$$\underbrace{|\mathbb{E}_{\rho^\pi,\tilde\pi}[Q_w - b_\phi^{\tilde\pi}] - \mathbb{E}_{\rho^\beta,\tilde\pi}[Q_w - b_\phi^{\tilde\pi}]|}_{(2)} +$$

$$\underbrace{|\mathbb{E}_{\rho^\beta,\tilde\pi}[(Q_w - b_\phi^{\tilde\pi})^- + \alpha\log\tilde\pi]|}_{(3)}\Bigg) \tag{84}$$

where

$$(1) \leq \mathbb{E}_{\rho^\pi,\tilde\pi}|Q^\pi - Q_w - (\mathbb{E}_\pi[(1+r_\phi)Q_w] - \mathbb{E}_{\tilde\pi}[(1+r_\phi)Q_w])|$$

$$\leq \Delta + 2D_{\mathrm{TV}}^{\max}(\pi||\beta)(1+K)M$$

$$\leq \Delta + \sqrt{2D_{\mathrm{KL}}^{\max}(\pi||\beta)}(1+K)M \tag{85}$$

$$\triangleq \Delta + \sqrt{2D_{\mathrm{KL}}^{\max}(\pi||\beta)}C_1$$

$$(2) \leq \sum_{t=0}^\infty \gamma^t\|\rho_t^\pi - \rho_t^\beta\|_1|\mathbb{E}_{\tilde\pi}[Q_w - b_\phi^{\tilde\pi}]|$$

$$\leq 2\Omega\sum_{t=0}^\infty \gamma^t t\sqrt{D_{\mathrm{KL}}^{\max}(\pi||\beta)} \tag{86}$$

$$\leq \frac{2\gamma\Omega}{(1-\gamma)^2}\sqrt{D_{\mathrm{KL}}^{\max}(\pi||\beta)}$$

$$(3) \leq \alpha C_\mathcal{H} + C_- \triangleq C_2 \tag{87}$$

Combining all parts, we have

$$|\eta(\tilde\pi - L_\pi(\tilde\pi))| \leq |\eta(\tilde\pi - \bar{L}_\pi(\tilde\pi))| + |\bar{L}_\pi(\tilde\pi) - L_\pi(\tilde\pi)|$$

$$= \frac{2\gamma\Upsilon}{(1-\gamma)^2}\sqrt{D_{\mathrm{KL}}^{\max}(\pi||\tilde\pi)} + (1-\omega)(\Delta + C_1\sqrt{2D_{\mathrm{KL}}^{\max}(\pi||\tilde\pi)} + \frac{2\gamma\Omega}{(1-\gamma)^2}\sqrt{D_{\mathrm{KL}}^{\max}(\pi||\beta)} + C_2) \tag{88}$$

## E  Action Bounds Transformation

The support of Beta distribution is $[0,1]$, for multivariate case with $n$ dimensions which are mutually independent, it would be $n$ products of $[0,1]$. In practice, the action bounds doesn't have to fit into this domain, we therefore need to apply a linear transformation upon each dimension to coincide with the actual bound $[m_i, M_i], i = 1, 2, \ldots, n$, where $m_i$ is the lower bound, and $M_i$ upper bound. It would be convenient to vectorize those bounds as $\mathbf{m}$ and $\mathbf{M}$.

Let $\mathbf{x} \in \mathbb{R}^n$ be a Beta-distributed random vector whose density is $f(\mathbf{x}|\mathbf{s})$. It is transformed as a new random vector $\mathbf{a} = x \cdot (\mathbf{M} - \mathbf{m}) + \mathbf{m}$ to be the action executed in the environment, at which product is element-wise. Denote $\mathbf{k} = \mathbf{M} - \mathbf{m}$ and $\mathbf{b} = \mathbf{m}$, its log-likelihood is given by

$$\log\pi(\mathbf{a}|\mathbf{s}) = \log f(\mathbf{x}|\mathbf{s}) - \mathbf{1}^\top\log\mathbf{k} \tag{89}$$

where $\mathbf{1}^\top$ is a $n$-dimensional vector whose entry is one. Since $\mathbf{1}^\top \log \mathbf{k}$ is a constant, the difference of the log-likelihood $\pi$ reduces to the that of the original one $f$, namely

$$\log \tilde{\pi}(\mathbf{a}|\mathbf{s}) - \log \pi(\mathbf{a}|\mathbf{s}) = \log \tilde{f}(\mathbf{x}|\mathbf{s}) - \log f(\mathbf{x}|\mathbf{s}) \tag{90}$$

This invariance is useful especially for the case that the log-likelihood policy gradient is involved e.g. A2C, TRPO and PPO. And it should be noted that only the $f(\mathbf{x}|\mathbf{s})$ is parameterized as $f_\theta(\mathbf{x}|\mathbf{s})$, thus $\nabla_\theta \log \pi(\mathbf{a}|\mathbf{s}) = \nabla_\theta \log f_\theta(\mathbf{x}|\mathbf{s})$. We store the transformed action $\mathbf{a}$ and the untransformed log-likelihood $\log f(\mathbf{x}|\mathbf{s})$, which is used in the policy improvement phase, to either calculate the difference of the log-likelihoods (TRPO, PPO) or the log-likelihood only (A2C). By detransforming the action $\mathbf{a}$ to the original one $\mathbf{x}$, $\tilde{f}(\mathbf{x}|\mathbf{s})$ can be calculated for those aforementioned purposes.

## F  Experimental Details

### F.1  Figure 2

All the on-policy algorithms are implemented from OpenAI Baselines Dhariwal et al. (2017), including A2C, TRPO and PPO. It is implemented with the latest version of SAC with auto-temperature Haarnoja et al. (2018b), from the author's repository[3], TD3[4] and IPG[5] as well. We perform all algorithms on the MuJoCo tasks with v3 version, except PPO, which is on the v2 version. The reason is that we observe the OpenAI Baselines implementation doesn't learn on the v3 version tasks at all, to make the results comparable, and respect the originality of the specific algorithm implementation, we have to make this compromise.

### F.2  Figure 3

The time complexity comparison was benchmarked with a single core on an Intel Xeon Platinum 8358 CPU. And the relative policy updates were calculated by dividing the total number of policy gradient updates by the total number of updates.

### F.3  Figure 4

The fixed interval is comprised of 2048 steps. We log the metric for 25% even spaced data within every interval.

**Variance of Policy Updates**   We monitor two consecutive time steps' policy parameters $\theta_t$ and $\theta_{t+1}$, and compute the policy change $\Delta\theta_t = \theta_{t+1} - \theta_t$, with which we have

$$\text{VPU} = \mathbb{V}[\Delta\theta_t] \tag{91}$$
$$= \mathbb{E}[\|\Delta\theta_t - \mathbb{E}[\Delta\theta_t]\|^2] \tag{92}$$

**Average Total Variation**   We define the average total variation as the sample average of the absolute differences between adjacent sample points

$$\text{ATV} = \frac{1}{N} \sum_{i=0}^{N-1} |\mathcal{L}_{i+1}^{\text{MSE}} - \mathcal{L}_i^{\text{MSE}}| \tag{93}$$

where $\mathcal{L}^{\text{MSE}}$ is the critic MSE loss. For PPO and SAC, it is the same as the critic loss, whereas we re-evaluate for DPO as it employs KL divergence loss.

---

[3]https://github.com/rail-berkeley/softlearning
[4]https://github.com/sfujim/TD3
[5]https://github.com/rlbayes/rllabplusplus

### F.4   Figure 5(a)

The calculation of trajectory variance is borrowed from Tucker et al. (2018), but with a tighter statistics.

We first follow the current policy $\pi$ to collect a large batch of data with a size of 25000, from which we uniformly draw a subset of data $\mathcal{D}_{\text{test}}$ with a size of 1000 as an estimation of the outer expectation. To estimate the trajectory variance, for each $(s, a) \in \mathcal{D}_{\text{test}}$, we independently sample two trajectories $\tau, \tau' \sim \tau|s, a$ with a maximum horizon 1000, with which we have a single variance estimator

$$\|u_\theta(s, a)\|^2 \left(\hat{A}(s, a|\tau)^2 - \hat{A}(s, a|\tau)\hat{A}(s, a|\tau')\right) \tag{94}$$

where $\hat{A}$ can be either $\hat{A}^{\text{UAE}(\gamma, \lambda)}$ or $\hat{A}^{\text{MC}}$. In the end, we average all the estimators and take log scale for better readability.

The reason we don't compare with GAE is that

- GAE is a strict special case of UAE.
- The critic of our algorithm is $Q_w$, for which GAE is no longer applicable.

### F.5   Figure 5(b)

We estimate both on- and off-policy gradient variance. In on-policy case, we perform the estimation on the collected rollouts (with a total steps 2048)

$$\frac{1}{|\mathcal{B}|} \sum_{(s,a)\sim\mathcal{B}} \|u_\theta\|^2 (Q_w - b_\phi^\pi)^2 \tag{95}$$

And for the case of off-policy, we uniformly draw a subset of states $\mathcal{D}_{\text{off}}$ (size of 10000) from the replay buffer, then sample a new action $a \sim \pi_\theta$ for each state. The estimator is

$$\frac{1}{|\mathcal{D}_{\text{off}}|} \sum_{s\sim\mathcal{D}_{\text{off}}, a\sim\pi_\theta} \|u_\theta\|^2 (Q_w - b_\phi^\pi)^2 \tag{96}$$

### F.6   Figure 5(c)

In our setting, the maximum horizon $T = 2048$, and the size of the replay buffer $|\mathcal{D}| = 1e6$, we therefore gradually increase the number of samples training the baseline starting from totally on-policy ($N = 1 \cdot T$) to the full samples ($N = |\mathcal{D}|$).

### F.7   Figure 6

We borrow the metrics from Fujimoto et al. (2022) with modifications. The mean absolute error (MAE), namely generalization error, is performed on the subset $\mathcal{D}_e$ of the test data $\mathcal{D}_{\text{test}}$ (will be introduced later)

$$\frac{1}{|\mathcal{D}_e|} \sum_{(s,a)\sim\mathcal{D}_e} |Q_w(s, a) - Q^\pi(s, a)| \tag{97}$$

where the test data $\mathcal{D}_{\text{test}}$ is collected by executing current policy $\pi$ in the environment with 50000 steps, and $\mathcal{D}_e$ is formed as the uniformly sampled data (size = 1000) from the test data. For each $(s, a)$ in the $\mathcal{D}_e$, the true action-value function $Q^\pi$ is approximated by the Monte-Carlo estimate with 100 episodes, each of which has a timelimit 1000 (which means each episode cannot exceed that limit). The root mean squared error (RMSE), namely prediction error, is performed on the data of the batch $\mathcal{B}$ (size = 2048)

$$\frac{1}{|\mathcal{B}|} \sum_{(s,a)\sim\mathcal{B}} \sqrt{\left(Q_w(s, a) - (r + \gamma Q_w(s', a'))\right)^2} \tag{98}$$

The $Q_w$ all of above is the mean value of the learned distribution $Q_w(s, a) = \mathbb{E}[Z_w(s, a)]$.

### F.8 Table 1

**DPO(A2C) implementation**  A2C is not a trust region method, therefore we only perform the policy update once.

**DPO(TRPO) implementation**  TRPO finds a search direction $\Delta\theta$ by the conjugate gradient algorithm with a backtracking line search. Though the interpolation is not straightforward at the first glance, we propose to firstly calculate off-policy loss $(1-w)\mathcal{J}_{\text{off policy}}$ under the current policy $\pi_\theta$, then search for an increment of $w\Delta\theta$. By parallelogram law, we can increment the policy parameter with $w\Delta\theta$, and then update the policy based on the pre-calculated off-policy loss.

And at each policy update, we update the distributional critic 10 times upon the minibatch drawn from $\mathcal{B}$.

### F.9 Table 2

| Setting | Description |
|---|---|
| **First group** | |
| no-UAE | replace UAE with Monte-carlo estimator |
| no-RB | set r $\equiv$ 0 |
| no-DIST | replace distributional critic and KL divergence with a single Q and MSE respectively |
| **Second group** | |
| no-INT | set $\omega = 1$ |
| no-ENT | set $\alpha = 0$ |
| **Third group** | |
| only-ON | without off-policy evaluation at the interaction level |
| only-OFF | without on-policy evaluation at the batch level |

## G  Implementation Details

### G.1  Network Architecture

Majority of the DRL algorithms parameterize the policy as a Gaussian distribution, due to its simplicity. However, the boundary effect of it is observed in several works Chou et al. (2017) Fujita & Maeda (2018). And in practice, the action space is usually bounded, thus it is beneficial to parameter our policy as a beta policy. Like Chou et al. (2017), the shape parameters $\alpha, \beta$ is forwarded through a fullly connected neural network, and converted to the range $[1, \infty)$ by a softplus activation added with a constant 1. For the critic, we parameterize it as a Gaussian distribution, output the mean value through the last layer, and the standard deviation operated by a softplus activation similarly. For the baseline, it is simply a fully connected neural network that outputs the residual term, when approximating the baseline, a constant 1 is added (see Equation 16). All the networks are with 2 hidden layers, and `tanh` activation function, each of which has 256 neurons. This architecture is robust to the hyperparameter changes, especially when different on-policy learners are involved.

### G.2  Advantage Interpolation

We introduce another interpolating parameter $\nu$ for advantage estimate over $\mathcal{B}$ before the policy improvement. Since we evaluate at each environment step, we expect a good quality that the expected value $Q_w = \mathbb{E}[Z_w]$ would achieve.

$$\hat{A} = (1-\nu)A^{\text{UAE}} + \nu(Q_w - b_\phi) \tag{99}$$

### G.3 Normalization

Unlike PPO, we minimize the code-level optimization, such as network initialization, learning rate decay, gradient clipping etc. Like PPO, we adopt observation normalization, reward normalization, and advantage normalization (batch level), as it is beneficial to stabilize the behavior of the neural network and enable faster learning.

## H Hyperparameters

| Hyperparameter | Value |
|:---:|:---:|
| optimizer | Adam Kingma & Ba (2015) |
| learning rate | $3 \times 10^{-4}$ |
| size of replay buffer $\mathcal{D}$ | 1000000 |
| size of mini batch from $\mathcal{B}$ | 256 |
| size of mini batch from $\mathcal{D}$ | 256 |
| discounted factor $\gamma$ | 0.99 |
| UAE $\lambda$ | 0.95 |
| target smoothing parameter $\tau$ | $5e-3$ |
| policy interpolating parameter $\omega$ | 0.7 |
| advantage interpolating parameter $\nu$ | 0.3 |
| temperature $\alpha$ | 0.03 |
| num samples | 30 |
| critic samples | 25 |
| **on-policy learner** | **PPO** |
| clipping parameter $\varepsilon$ | 0.2 |
| size of batch $\mathcal{B}$ | 2048 |
| epochs per batch | 10 |
| baseline updates | 12 |
| **on-policy learner** | **A2C** |
| size of batch $\mathcal{B}$ | 256 |
| epochs per batch | 1 |
| baseline updates | 4 |
| **on-policy learner** | **TRPO** |
| max kl | 0.1 |
| damping | 0.1 |
| size of batch $\mathcal{B}$ | 4096 |
| epochs per batch | 1 |
| baseline updates | 12 |

Table 3: Hyperparameters of DPO

# I Additional Experiments

## I.1 Variants of DPO

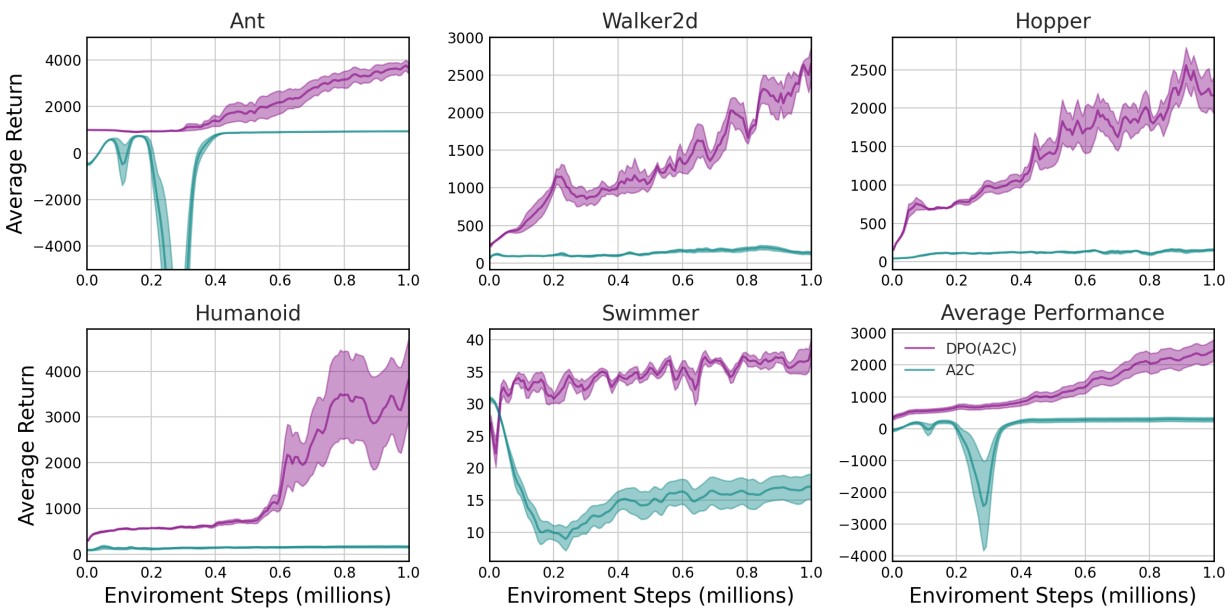

Figure 7: Comparison between DPO(A2C) and A2C, averaged over 5 random seeds and shaded with standard error.

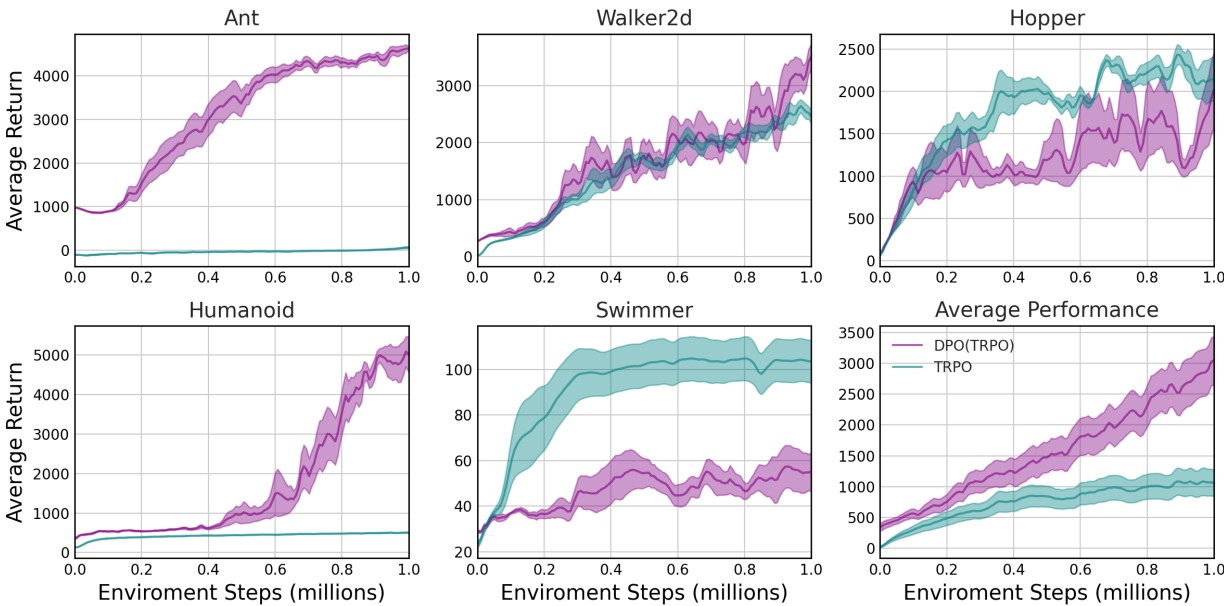

Figure 8: Comparison between DPO(TRPO) and TRPO, averaged over 5 random seeds and shaded with standard error.

## I.2 Additional Baselines

Table 4: Comparison to other algorithms combining on-policy methods with off-policy data.

| METHOD | WALKER2D | HOPPER | SWIMMER | HUMANOID | ANT | AVG. |
|---|---|---|---|---|---|---|
| DPO (1M) | **4860** ± 680 | **3187** ± 351 | **112** ± 13 | **6285** ± 852 | **5278** ± 173 | **3944** ± 414 |
| Q-PROP (1M) | 358 ± 26 | 1464 ± 203 | 59 ± 4 | 355 ± 3 | -46 ± 8 | 438 ± 49 |
| P3O (3M) | 3771 | 2334 | - | 2057 | 4727 | 3222 |

Results for Q-prop were replicated by following the official repository, while P3O's results are based on the paper.

## I.3 Effect of Parameters

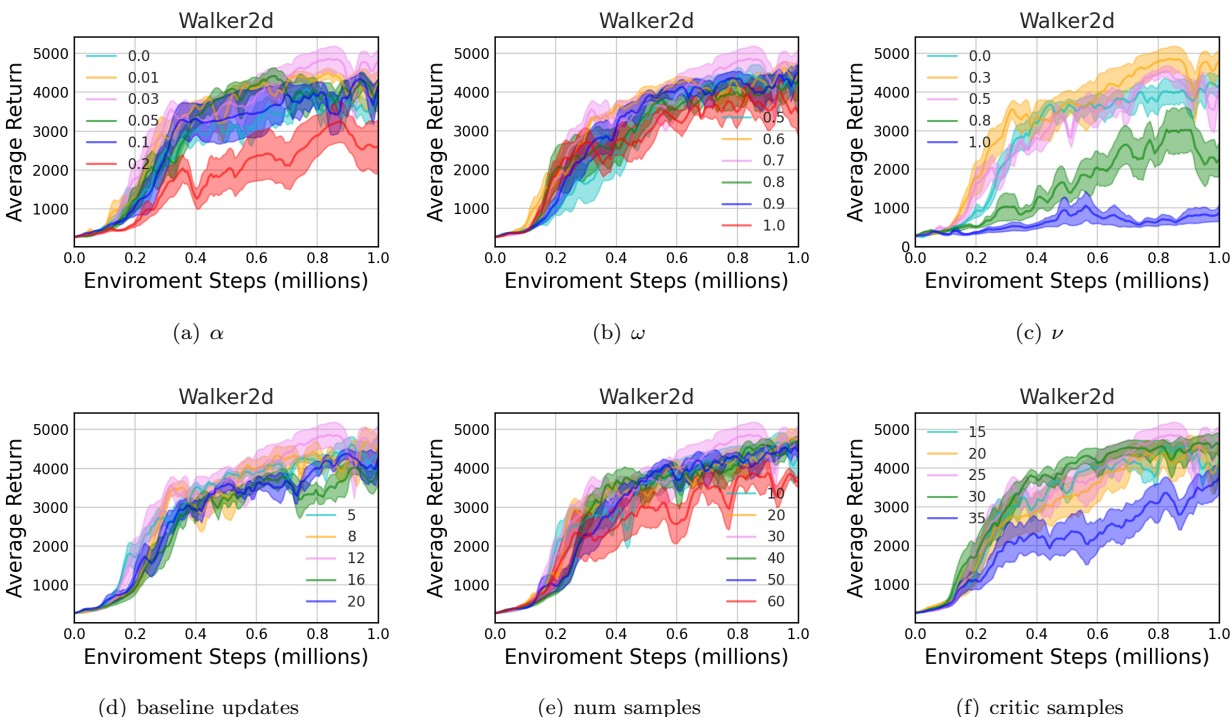

Figure 9: Varying levels of different parameters of DPO: **(a)** temperature $\alpha$, **(b)** interpolating parameter $\omega$, **(c)** advantage interpolating parameter $\nu$, **(d)** baseline updates, **(e)** num samples and **(f)** critic samples.

## I.4   Validation

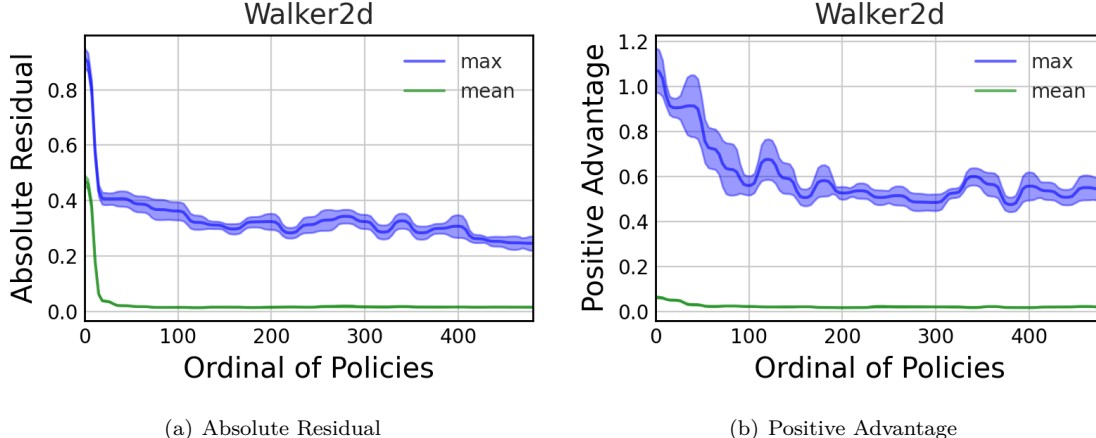

(a) Absolute Residual                                    (b) Positive Advantage

## I.5   Wall Clock Time

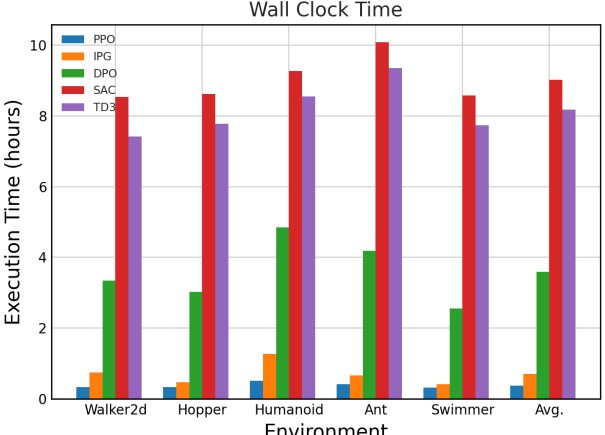

# J   A New Learning Paradigm

**General**   DPO demonstrates robustness across various on-policy learners and the majority of hyperparameters.

**Lightweight**   DPO boasts an impressive reduction in training time, requiring only 4% of the policy gradients when compared to off-policy algorithms like TD3 and SAC.

**Parallelable**   DPO can efficiently utilize multiple environments to collect samples like PPO, improving training efficiency and reducing training time.

**Sample Efficient**   DPO fully exploits two sources of data for both evaluation and control, greatly boosting the sample efficiency of on-policy algorithms.

**Stable**   DPO stands out for its stability, characterized by smoother policy changes and reduced variability in critic MSE loss, ensuring more reliable learning.

