# OpenReview forum: "Distillation Policy Optimization"
_TMLR — Rejected by TMLR_

### Review · Reviewer_YEzR · 2023-10-11

**Summary Of Contributions:**

The paper aims to propose a new combination of on-policy and off-policy algorithmic updates, in order to combine the strengths of both paradigms. To do this, the paper proposes a number of changes: a new advantage estimator, a new way to construct baseline and a few experimental setups which arguably improve the performance of the downstream algorithms. The paper shows modest gains in certain continuous control tasks.

**Audience:**

Yes

**Broader Impact Concerns:**

No.

**Claims And Evidence:**

No

**Requested Changes:**

I have a few questions that I hope the authors can address during rebuttal.

=== **distributional RL** ===

The idea of distributional RL is orthogonal to combining on and off-policy updates for deep RL. In that sense, I find it difficult to understand the motivation to mention distributional RL in Sec 2.4 and as an experimental setup in Sec 5.1. Removing the distributional nature of the update, will we see the same benefits and phenomenon? How does the fact that the authors consider distributional updates impact the final results and the theoretical conclusion, and how do results differ from the mean-based RL case.

To me, introducing distributional RL deviates the paper a bit from the core idea it tries to present, and looks a bit confusing.

=== **UAE** ===

It seems that the main contribution of UAE compared to GAE is to introduce another term $z_t=\Phi_t-b_t$ over time into the estimate. However, are these terms zero-mean? That is, do we have $\mathbb{E}[z_t|s_t]\approx 0$ in general? If not, this can introduce bias into the estimate. Indeed, since $\Phi_t$ can be $V(s_t),Q(s_t,a_t)$ and both are function approximations, the terms are in general not zero-mean. Or maybe the authors mean that $\Phi_t$ is always a Monte-carlo estimate of future returns, in which case $z_t$ can still be non-zero-mean because $b_t\neq V^{\pi}(s_t)$. However, in this latter case this does not introduce bias into the PG update.

Also, the new estimate indeed looks more general, but why is this desirable? Do we expect the variance to be reduced further even when not considering the bias effect? GAE is already a sum of almost zero-mean TD error, and the variance reduction is carried out by the value function baseline. What's the extra benefit of having $z_t$?

Hope that the authors can clarify further on the technical details here.

=== **residual baseline** ===

It is not clear to me why it is important to approximate the "optimal baseline", since in practice learning value function $b\approx V^\pi$ is almost good enough to reduce variance of the policy gradient estimate.

Meanwhile, theoretical considerations aside, the technical details of Sec 4 are not clear to me. For example, does the loss Eqn 13 mean that we update both $Q_w$ and $b_\phi$ with the loss function, or is there a stop-gradient somewhere? Does Eqn 12 refer to a new parameterization with $r_\phi$, so that we can update $r_\phi$ to learn $b$?

=== **Sec 5.1** ===

As discussed above, what's the importance of having distributional regression here instead of regular regression? Distributional updates introduce further complexities into the discussion, and in addition, all theoretical and algorithmic discussions thus far have focused on non-distributional setting. It feels abrupt to see experimental setups in Sec 5.1, without a comparison to the mean-based setting at least.

And then further down in Sec 5.2, we are back to the mean-based RL case. Notations such as $A^{\pi_\theta,V}$ are not introduced properly.

=== **Thm 6.9** ===

Performance guarantee is still largely based on the performance difference lemma and results from trust region policy optimization, as evidenced by the KL divergence. Though this is almost inevitable, the bound itself can be too loose for practical situations, because off-policy data can differ a lot from the learned policy, in which case the KL divergence becomes meaningless for bounding the actual performance. It is not quite feasible to characterize the off-policy updates and performance guarantee using KL-divergence and trust region arguments, because it intuitively goes against the idea of doing "off-policy" updates.

=== **Related work**===

Sample efficient actor-critic with experience replay by Wang et al, 2016 is a related reference that aims to combine on and off-policy data.

=== **Experiment ** ===

The performance gains in Fig 2 are not very strong to me, as DPO performs on par with other baselines. Though gains in Humanoid are pretty significant, it is not clear if other algorithms can be boosted with more hyper-parameter tuning (in all fairness, the new algorithm introduces a number of hyper-parameters to tune).

When DPO compares against A2C and TRPO, is DPO allowed to utilize off-policy data from past buffers?

Since the theoretical contributions are a bit weak, one would expect a more solid empirical gain of the algorithm.

**Strengths And Weaknesses:**

The strengths of the paper lie in that it has investigated an interesting problem of combining off-policy and on-policy updates for deep RL. The weaknesses mainly consist in that the contributions of the paper are not super clear to me. The paper has proposed a few seemingly interesting ideas, feeling like "a bag of tricks", but how they come together into a single unified framework is not clear to me. The theoretical and empirical contributions are also relatively weak.

---

### Review · Reviewer_YHfD · 2023-10-14

**Summary Of Contributions:**

This paper proposes DPO, a method for combining on-policy and off-policy learning. This algorithm combines several changes on top of a base on-policy learning algorithm like PPO including: unified advantage estimator (a modification of GAE), a method for learning a baseline function, distributional RL, and an optimistic objective. The paper presents some theory to support the method and then experiments on mujoco tasks showing data efficiency that matches off-policy algorithms.

**Audience:**

Yes

**Claims And Evidence:**

Yes

**Requested Changes:**

1. Can you include either (a) results for the baselines with similar modifications as DPO (distributional RL, optimistic objective) or (b) results for DPO compared to the baselines without these modifications? This would help clarify the comparison to baselines on more equal footing.

2. Can you select the most important theoretical results, make their statements a bit more clean, and explain the consequences more fully? Other results and lemmas can be moved to the appendix, but this would make the theory section more useful to the reader.

3. Can you respond to the concerns raised above about particular theory results?

**Strengths And Weaknesses:**

### Strengths

1. The paper presents some potentially interesting algorithmic ideas about generalizing the GAE to use a baseline that is learned off-policy. This could lead to a method that combines the benefits of on-policy and off-policy learning.

2. The empirical results on mujoco tasks are fairly strong, showing performance that is competitive with standard off-policy algorithms.

3. There are some interesting experiments suggesting that the method does have lower variance than standard off-policy methods, so it may capture some benefits of on-policy algorithms.

### Weaknesses

1. The paper proposes several algorithmic changes at once, making it difficult to properly compare to some baselines. In particular, on top of the novel contributions of UAE and residual baselines, the paper uses distributional RL and an optimistic objective. These modifications are not the main contribution of the paper, but they are not present in the main baselines considered (e.g. SAC, PPO, etc). As a result it is not quite clear that the main experimental results are making clear comparisons between off-policy, on-policy and mixed settings, because orthogonal algorithmic choices like distributional RL are not held constant across the various settings. The paper does include ablations that make the argument that removing components of the algorithm hurts performance, but some of the changes (e.g. distributional RL) are somewhat orthogonal to the novel contributions and are not present in the baselines.

2. Experiments are only presented on simple mujoco control problems. Extensions to more complicated problems beyond locomotion would provide a stronger argument for the method.

3. The theory section has a few issues. First, is too dense and does not yield clear takeaways to the reader. Moreover, the motivation and assumptions for the particular results are often unclear. On Corollary 6.5, $ \Psi $ is taken to be $ V$, but in the algorithm, $ \Psi $ is taken to be $ Q$. This discrepancy seems to make the theory not applicable to the actual algorithm. Also, in Corollary 6.5, the assumption that $b $ reduces variance more than $ V$ sort of seems to presume the desired result. In corollary 6.7 it is not clear why the variance of the Q function under the replay buffer distribution under a successor policy is an important quantity of interest. It seems like the important thing is whether the baseline is reducing variance under the distribution induced by the current policy to reduce policy update variance (as in Eq 17). Also, in corollary 6.7, the result does not seem very tight since the factor 2 in front of J seems unnatural (especially consider the case where $ \epsilon = 0$). It is unclear what the point of Theorem 6.8 is since it already assumes that policies converge to optimal and residuals converge to 0, it is not surprising that advantages must also converge to 0. The assumptions here are so strong that the conclusion seems direct, but maybe I am missing some subtelty.

---

### Review · Reviewer_M8id · 2023-11-07

**Summary Of Contributions:**

This work introduces a new general RL algorithm, DPO, which mixes several improvements over known off-policy and policy methods. The methods contains thee improvements: a new way of using replay data, with an off-policy loss for the q-value and baseline; a new way of estimating the advantage; and using a distributional critic.

DPO is evaluated on standard mujoco benchmark, and proves to be efficient, in particular in terms of sample efficiency.

**Audience:**

Yes

**Broader Impact Concerns:**

No concerns.

**Claims And Evidence:**

No

**Requested Changes:**

I think two changes arise from the weaknesses section of my review:

1. Overall, I think the organisation of the paper could be improved, notably by tackling issues covered in the Clarity section. In particualr, I think situating the use of distributional RL in the context of the work and reworking Section 6 are the more important changes for clarity. While not crucial, I would also appreciate an explanation for the title somewhere in the paper.

2. The experiments suggested in the Results section would be very useful (ablation of the distributional critic, SAC with more replay steps). Note that I checked "No" in "Claims and evidence" because of that (I don't think we can conclude the usefulness of DPO without the abaltion of the distributional aspect).

**Strengths And Weaknesses:**

# Strengths

Sample efficiency is a key  and useful problem to tackle for RL; this line of work could be very beneficial to the community.

DPO shows convincing improvements in terms of sample complexity.


# Weakness

In my my opinion, there are two main weaknesses in this work. The first is the clarity and presentation of the work, and the other is the quality of the results.

## Clarity

In general, I found the paper quite hard to follow, and not super clear. I think the organisation and presentation of ideras coulld be imporved on several points, that I detailed below. Note that these remarks are quite general and apply throughout the paper, I will give examples to clarify my concerns.

**Motivations**

One of the main weaknesses of the work is that the motivation and contributions of the work are not always clear. The main contribution is a new algorithm, which uses
 - A replay buffer together with on-policy elements
 - A new advantage estimator (UAE and the new baseline)
 - A distributional critic

From the paper, it is not obvious to me why one would want to study these three aspects together.  I think it arises because the motivation for the work is not super clear. I understand from the introduction that it boils down to "get the best of both world in on-policy and off-policy", but I think it lacks some precision on what you what to achieve.

Furthermore, I don't think the link with distributional idea is at all discussed. This contribution is a bit of problem in my opinion: it looks like using a distributional critic is not really related to the motivation of the paper (using both on-policy and off-policy data); and the algorithm is only compared to baselines that do not use a distributional critic.



**Organisation**

I think this work can be confusing at times because it introduces many elements but the organization of the paper makes it hared to understand how and why they are connected.

*Examples of what I mean*

- The distributional critic is introduced in preliminaries, but we only learn in Section 5 that it is actually used by DPO. It is also not listed in contribution section of the introduction. Also, it is not clear in the presentation if the method is new or is taken from a previous paper.

- Same thing with the optimal baseline: we only understand why it was introduced later, in Section 4. Also, in 2.3, it is a bit confusing to define an optimal baseline without stating according to what criterion it is optimal.

 - UAE is introduced in Section with very little context regarding why it its useful, we only understand the optimal baseline after.

 **Title**

I do not understand the title of the paper: why "distillation" ? (this is a bit of detail, but actually quite confusing that the title is not reflected in the abstract).

**Theoretical analysis**

Section 6 lacks quite a lot of discussion in my opinion for two reasons.

First, t starts with three questions, but then the result are not explicitly linked to answers to these questions.

Second, it is not obvious to me what the point of the analysis is here. It is not really linked to the practical method, or it lacks significant discussion to explain why you think this kind of analysis is useful or necessary.

## Results
I think some aspects of the experimentation could be improved.

**Distributional critic**

The distributional critic should be ablated. D-SAC [Duan et al, 2021] is very strong, especially on Humanoid, so I am worried it could explain a large portion of the performance.

 **Comparison to SAC with more replays**

A key experiments would be SAC with a comparable ration of gradient updates per env steps. The choice in default SAC is arbitrary, and since the loss is off-policy,  in principle you could use a much higher ratio of gradient steps per environment steps. It is possible that jut seeing again the data several times could increase the sample efficiency  of SAC.


**Experimental policy**

What do you mean by  "We thus evaluate our algorithm by executing the mean action with 10 trails" ?

## Minor comments

- Citations are not formatted properly

- The cite for "replay buffers" is a bit too recent. Min et al actually cite [Long-Ji Lin, Reinforcement learning for robots using neural networks. Technical report, DTIC Document, 1993] for the replay mechanism.

- The term "deadly triad" dates back before van Hasselt 2018. I found earlier occurrences by Sutton in as Neurips tutorial in 2015 ( http://www.incompleteideas.net/Talks/RLtutorialNIPS2015.pdf); and the notion was actually already present in the 1998 version of his RL book (although the term "deadly triad" is not used  ) ( http://incompleteideas.net/book/ebook/node92.html ).

---

### Decision · Action_Editor_X5y5 · 2024-01-10

**Recommendation:** Reject

**Comment:**

My reasoning for recommending a reject is that I think the paper's clarity still needs a substantial improvement. All of the reviewers have highlighted that this is an issue, and while the authors have addressed some of the points raised, I do not feel the paper is ready for publication yet. In particular, while the paper's abstract starts with a clear problem (combining off- and on-policy data), the paper ends up being a combination of various techniques that don't clearly stem from the same motivation.

For instance, while extended GAE to UAE is a valuable exercise, it's connection to the stated problem (combining off- and on-policy data) is less clear. Inded, it seems like this contribution is more general and could be applied to any other method that benefits from variance reduction techniques. The same comment holds for the distributional critic: it seems like a generally useful thing but not directly stemming from the original motivation.

Thus, while the combination of these techniques seem to produce a reasonably good agent, it is not clear why these particular designs were chosen; could we not have used other techniques for variance reduction and sample efficiency? As discussed in the Claims section above, the ablations performed on the used components are not convincing enough to justify their need.

For these reasons, I am recommending a rejection, with an invitation to resubmit with a clearer (and more streamlined) algorithmic proposal. The ideas presented in the paper are interesting and could be of value to the community, they just need to be presented in a clearer, and more statistically justified, manner.

**Audience:**

Yes.

**Claims And Evidence:**

The ablations showed in Table 2, which presumably justify the combination of all these tricks, are not convincing enough to justify their need. In particular, most of the confidence intervals between the ablations and the full DPO performance overlap, which suggest that the difference is not significant. This calls into question the necessity to have all of them combined into a single agent, which is one of the main sticking points for all reviewers.

**Resubmission Of Major Revision:**

The authors may consider submitting a major revision at a later time.